# AutoClinician: Structured Clinical Guideline Integration for Trustworthy Diagnostic Reasoning in Healthcare

## Abstract

Recent advances in large language models (LLMs) have enabled clinical conversational systems with impressive diagnostic capabilities. However, existing approaches often lack alignment with real-world clinical workflows and fail to provide interpretable, evidence-grounded reasoning. In this work, we propose AutoClinician, a unified and training-free framework that integrates clinical guidelines to support stepwise and explainable diagnosis on real-world electronic health records (EHRs). AutoClinician first extracts and summarizes narrative guidelines into Clinical Evidence Graphs (CEGs). These graphs are further automatically verified and refined using a consistency-based strategy. To support trustworthy and patient-specific diagnosis, we utilize CEGs by conducting context-aware, evidence-grounded clinical reasoning on EHRs with Deterministic Finite Automaton (DFA). Our framework outperforms both general-purpose and clinically specialized LLMs, and exhibits stronger interpretability. Code is available here [1].

## 1 Introduction

Recent advances in LLMs (Touvron et al., 2023; Wu et al., 2023) have created new opportunities to enhance diagnostic assistance and patient interactions in healthcare (Biswas, 2023; Li et al., 2023b; Shah, 2024; Singhal et al., 2023). Clinical conversational systems have demonstrated competitive performance in medical question-answering tasks (Wang et al., 2023a; Yang et al., 2024b). However, as shown in Figure 1, most existing efforts adopt a pure text-based setting that deviates significantly from real-world clinical workflows, focusing on tasks such as answering USMLE-style questions (Wang et al., 2023a; Labrak et al., 2024). These medical questions mainly address general inquiries but are limited in supporting clinical workflows such as test recommendation, result interpretation, and follow-up planning. In real-world clinical workflows, patient clinical events are documented in Electronic Health Records (EHRs) (Shickel et al., 2017), which have been largely overlooked in recent work. Moreover, high performance alone is insufficient for clinical deployment. Physicians require AI assistants that support evidence-grounded reasoning, where each decision adheres to the principles of *evidence-based medicine (EBM)* to ensure transparency, legal defensibility and clinical accountability (Subbiah, 2023).

Extending LLMs to work with EHRs is critical for aligning them to real-world medical workflows. However, the design of EHR-enhanced clinical conversational agents faces key limitations: (1) **Difficulty in interpreting medical records**. EHRs contain numerical and domain-specific data that require precise clinical interpretation, posing fundamental challenges for text-based LLMs which lack grounding in the diagnostic significance of various laboratory test values (Li et al., 2024). Recent studies show that GPT-4's diagnostic accuracy drops

---

[1] https://anonymous.4open.science/r/Autoclinician-0AAD

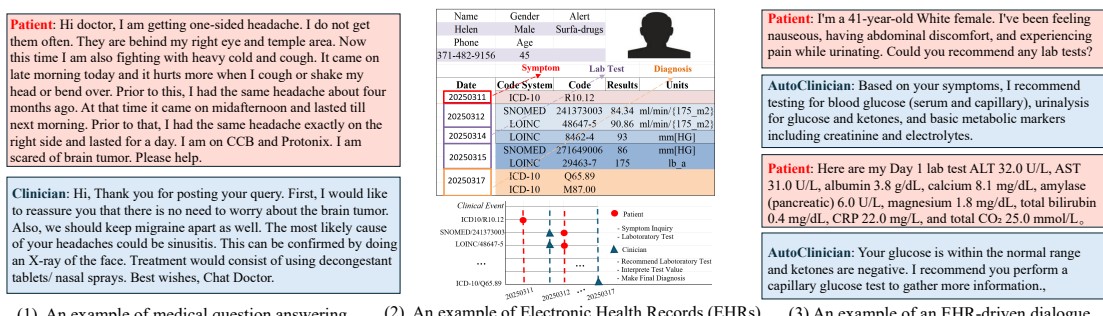

(1) An example of medical question answering.    (2) An example of Electronic Health Records (EHRs).    (3) An example of an EHR-driven dialogue.

Figure 1: Comparison between classical medical QA and EHR-driven diagnosis. (1) A typical medical QA setting, which takes a textual question and directly returns a diagnosis result. (2) Electronic Health Records (EHRs), which document patient information such as symptoms and laboratory test results, and are routinely used in real clinical workflows. (3) AutoClinician performs guideline-grounded reasoning on EHRs to generate diagnoses with explicit rationales.

significantly when test results are not properly interpreted or integrated (Bhasuran et al., 2025). (2) **Lack of explicit medical rationale.** There are underlying rationales behind EHRs, such as why a specific test was requested or how the diagnosis is supported by test results. Capturing this reasoning evidence is essential for building trustworthy clinical assistants, yet it has been overlooked by existing work. To address these challenges, we pose a critical research question: how do we empower foundational LLMs with expert medical knowledge and evidence-grounded clinical reasoning abilities?

To overcome these limitations, we propose to incorporate *official clinical guidelines* into the reasoning of diagnostic agents. It is known that official clinical guidelines establish evidence-based diagnostic criteria, define validated thresholds for clinical interpretation, and provide legal justification for diagnosis. However, automating clinical guideline integration for diagnostic conversational systems faces three key challenges. (1) **Narrative complexity**. Clinical guidelines are often lengthy and comprehensive. It is important to extract concise and discriminative evidence to assist LLMs with diagnostic workflows. (2) **Lack of guideline-grounded validation mechanism**. The absence of concise guideline knowledge makes it difficult to evaluate the correctness of extracted information. Human evaluation or expert-annotated supervision is prohibitively costly for its scalability. (3) **Trustworthy reasoning for each patient**. Designing a transparent reasoning paradigm that is grounded in guideline-supported rationales is critical for a trustworthy diagnostic agent.

Concretely, we propose **AutoClinician**, a training-free framework that leverages clinical guidelines to reason over EHR-based dialogues. AutoClinician comprises two key components: **(1) Clinical Guideline Compression and Refinement**. We introduce Clinical Evidence Graphs (CEGs), which are distilled from clinical guidelines to encode structured diagnostic evidence. To ensure accuracy and scalability, we propose a consistency-based strategy that generates pseudo–patient examples to automatically refine CEGs quality at scale. **(2) Conversational Reasoning via Patient-Specific Deterministic Finite Automaton (DFA)**. We model diagnostic interactions using DFA. This structured state tracker integrates an LLM with Retrieval-Augmented Generation (RAG), enabling an interpretable and stepwise reasoning process.

The contributions of our work are listed as follows: (1) We develop AutoClinician, the first framework to integrate clinical guidelines into EHR-based diagnostic conversations, enabling evidence-grounded reasoning. (2) We propose a scalable, unsupervised algorithm that automatically refines CEGs from clinical guidelines through a consistency-based strategy. (3) We introduce a DFA to track and manage patient-specific reasoning based on CEGs, enabling evidence-supported conversations. (4) AutoClinician outperforms both general-purpose and medical-specialized LLM baselines in laboratory test recommendation and diagnosis prediction, demonstrating superior reasoning accuracy, interpretability, and adherence to clinical guidelines.

## 2 RELATED WORKS

**Diagnosis Modeling from Electronic Health Records**. EHRs encode diagnostic trajectories through structured data such as demographics, symptoms, and clinical test results, making them a key resource for diagnosis prediction. One research direction focuses on enriching medical concept representations using external knowledge sources like clinical ontologies and knowledge graphs (Choi et al., 2017; Panigutti et al., 2020; Jiang et al., 2023; Wang et al., 2023b), but these often suffer from limited disease coverage and poor generalizability. To address this, recent methods employ unsupervised learning and LLMs to derive code embeddings via self-supervised relational graphs (Yao et al., 2024), contrastive pretraining (Cai et al., 2022), or code generation (Ma et al., 2024), yet they frequently overlook EHR heterogeneity and omit numerical laboratory test values. A second line of work treats EHRs as tabular data, framing diagnosis prediction as a supervised classification task. Methods such as XGBoost (Chen & Guestrin, 2016), generalized additive models (Hastie, 2017), piecewise linear functions (Montomoli et al., 2021), and rule-based learners (Ren et al., 2024) aim to learn decision functions over laboratory test values. However, these models are prone to population bias and often lack alignment with clinically grounded criteria.

**Retrieval-Augmented Generation for Medical Dialogue Systems.** Integrating external knowledge sources is critical for dialogue systems to generate trustworthy responses. In biomedical domains, Self-BioRAG (Jeong et al., 2024) dynamically searches documents from knowledge corpus to improve the response quality. MRD-RAG (Chen et al., 2025) further explores multi-disease retrieval by analyzing semantic relationships and contrasts among candidate diagnoses. CLEAR (Lopez et al., 2025) extracts information from clinical notes to improve the understanding of EHR data. RAICL (Zhan et al., 2025) combines retrieval-augmented generation with in-context learning to select disease-relevant demonstrations to improve image-based diagnosis performance. MAVEN (Jadhav et al., 2024) applies RAG to identify and correct factual errors in clinical notes. For a comprehensive overview, we refer readers to the recent survey (He et al., 2025). In contrast, we retrieve from clinical guidelines and refine compressed guideline graphs to support evidence-based diagnosis.

**Learning from Clinical Guidelines.** Learning discriminative structures from clinical guidelines is crucial for clinical decision support (Zhu et al., 2024; He et al., 2024). Early works developed rule engines (Mei et al., 2011) to construct rules from medical literature. Previous work formalizes guideline-driven decision processes as an information extraction problem. For example, previous work (Wu, 2022) combines a BERT-style encoder (Cui et al., 2021) to extract relation triples and compose structured trees. PromptRE (Jiang et al., 2022) frames tree extraction as a multi-round conditional relation extraction task. Recent works (Zhu et al., 2024; Li et al., 2023a; He et al., 2024) further leverage LLMs to transform unstructured medical texts into machine-executable decision trees. Another line of works first generate if-else pseudocode and then refines the trees (Hou et al., 2025). However, these methods primarily extract textual information, while neglecting numerical data that are critical for interpreting laboratory test results. Moreover, they mainly rely on tree-structured representations, which impose a single-parent hierarchy and thus fail to capture the concurrent and interdependent nature of real-world diagnostic logic.

## 3 METHODOLOGY

**Overview of Autoclinician**: In this work, we propose *AutoClinician*, the first framework that integrates clinical guidelines into diagnostic reasoning over EHRs via a retrieval-and-reasoning pipeline. AutoClinician automatically compresses guidelines into Clinical Evidence Graphs (CEGs) and employs a consistency-based strategy to validate and refine them (Section 3.1). Given a user query containing demographics and symptoms, AutoClinician retrieves relevant CEGs, and guides LLMs to generate responses through structured reasoning over these graphs.

## 3.1 LEARNING CLINICAL EVIDENCE GRAPHS FROM GUIDELINES

In this section, we first introduce the construction process and formal definition of CEGs, and then present a consistency-based strategy that synthesizes pseudo–patient examples from source guidelines to iteratively refine these graphs.

### 3.1.1 CONSTRUCTION OF CLINICAL EVIDENCE GRAPHS

First, we extract and summarize concise and critical rationales from guidelines in the form of Clinical Evidence Graphs (CEGs). Each graph encodes the reasoning pathways for one guideline, where each node is annotated with a `<ID, context, trigger, state, termination>` tuple. We define these node attributes as follows:

- **ID**: Denotes the index of current node in the graph.
- **Context**: Represents the current diagnostic status, (e.g., 'Context = Confirm Testing' after patient completing relevant laboratory tests). It links to the `state` of its parent node, ensuring consistent progression along the diagnostic pathway.
- **Trigger**: Specifies the clinical condition or threshold defined by guidelines. It is activated when patient's laboratory results or clinical observations meet a specified criteria. For example: 'Trigger = {A1C $\geq$ 6.5%, FPG $\geq$ 126 mg/dL, or random glucose $\geq$ 200 mg/dL}' indicates suspicion of diabetes.
- **State**: Specifies the guideline-prescribed clinical action taken when the `Trigger` is activated. It reflects how to act upon the observed condition based on clinical guidelines. For example, 'State = Suspect diabetes' when the patient's laboratory tests satisfy the defined `Trigger`.
- **Termination**. True or False. If true, this node is a leaf (final diagnosis/conclusion). If false, the engine will continue by locating the next step.

This structured 5-tuple formalizes the stepwise clinical reasoning encoded in guidelines, enabling systematic modeling of diagnostic pathways for downstream tasks. Concretely, we first collect and parse guidelines and use a state-of-the-art LLM to identify diagnosis-relevant sections. These identified sections are transformed into symptoms and corresponding diagnostic logic graphs, where symptoms serve as the key for CEG retrieval. Notably, each graph may lead to multiple diagnostic outcomes. For example, a diabetes-related guideline can terminate in different diagnoses, such as type 1 or type 2 diabetes [2]. Appendix B.2 provides the prompts for diagnosis-related section extraction, and Appendix B.3 provides the prompts for CEGs generation. A CEG derived from a diabetes-related guideline is shown in Appendix C.4.2.

### 3.1.2 AN AUTOMATIC VERIFICATION AND REFINEMENT STRATEGY FOR CEGS

CEGs abstract diagnostic rules from guidelines into structured decision paths, but their accuracy fundamentally depends on the capabilities of base LLMs. To ensure their correctness, a natural solution is to incorporate human-in-the-loop verification. However, given the extensive volume of guidelines and the cost of expert evaluation, this approach is inherently limited in scalability and susceptible to human bias and error.

To address these limitations, we design an automatic verification and refinement strategy that synthesizes pseudo–patient examples from the source guidelines and employs them to iteratively refine the CEGs. Intuitively, each pseudo-patient example can be viewed as an instantiation of a specific CEG branch, serving as a concrete test case of the abstract rule. Because the CEG and the pseudo-patient examples share a common source, any inconsistency (for example, a pseudo-patient example that fails the CEG's logic) pinpoints a

---

[2]https://diabetesjournals.org/care/article/46/Supplement_1/S19/148056/
2-Classification-and-Diagnosis-of-Diabetes

misalignment that need to be corrected. To systematically enhance the effectiveness of CEGs, we first examine recurrent error modes and subsequently construct pseudo-patient examples to address them.

Based on our analysis with professional clinicians, we first identify four common error modes in CEGs: (1) **Incomplete Evidence** that misses diagnostic conditions, such as subtle symptoms or laboratory tests. (2) **Threshold Misalignment** that misassigns numerical thresholds to incorrect laboratory tests. (3) **Logic Errors** that misorders diagnostic test steps specified by guidelines. (4) **Conflict Resolution Failure** that handles conflicting symptoms or laboratory tests. Detailed examples can refer to Appendix C.4.1.

To systematically prevent and correct error modes, we design four types of pseudo-examples tailored to specific failure cases. Each pseudo-example is assessed by an LLM-as-a-judge, producing a binary alignment score with respect to the generated CEGs. If the alignment score falls below a predefined threshold (0.8 in our experiments), the LLM is prompted with failed pseudo-patient examples to refine the initial CEGs. These flagged examples are incorporated as in-context demonstrations, guiding the LLM to recover missing constraints, adjust threshold logic, and revise invalid graph transitions. This strategy enables unsupervised validation and correction of clinical reasoning structures without expert supervision. The prompt used for refinement is provided in Appendix B.4.

## 3.2 MODELING CONVERSATIONS WITH A DFA-GUIDED STATE TRACKER

In this section, we model the diagnostic workflow as a Deterministic Finite Automaton (DFA), which aligns multi-step reasoning with clinical guidelines. The DFA acts as a conversational state tracker, summarizing the patient's current status, comparing EHR evidence with triggers of CEG nodes, and driving guideline-informed state transitions accordingly. This design provides a transparent and structured scaffold that ensures faithfulness between diagnostic reasoning and guideline logic.

### 3.2.1 CONVERSATIONAL STATE REASONING TRACKED VIA DFA

DFA is used to model structured decision processes (Minsky, 1956). Formally, a DFA is defined as a tuple $(\mathcal{S}, \Sigma, \delta, s_0, F)$, where $\mathcal{S}$ is a finite set of states, $\Sigma$ denotes the input space, $\delta : \mathcal{S} \times \Sigma \to \mathcal{S}$ is the transition function, $s_0 \in \mathcal{S}$ is the start state, and $F \subseteq \mathcal{S}$ is the set of terminal states. We capture the reasoning process by defining a DFA as the *Conversational State Tracker*:

- $\mathcal{S}$: The set of conversational states, each representing the evolving diagnostic context defined by accumulated patient observations and reasoning progress. Concretely, we specify five distinct states: {Start, PartialResults, ConflictResults, ConfirmedDiagnosis, Unresolved}, where $s_0 =$ Start and $F =$ {ConfirmedDiagnosis, Unresolved} are terminal states indicating completion of the reasoning process.

- $\Sigma$: The input space, which captures the output of diagnostic reasoning step after examining the EHR record with guideline CEGs. Specifically, we define five categories of inputs:

  - ConditionSatisfied: Given a patient's input (laboratory test results or symptoms), the evidence is compared against the conditions specified in the trigger of the current CEG node. If the conditions are met, the input is labeled as ConditionSatisfied, and reasoning proceeds to the corresponding child node in the next step. For example, it will evaluate whether a laboratory value is abnormal according to a guideline-defined threshold, or whether multiple tests jointly satisfy a logical condition such as "A is abnormal *and* B is abnormal."

  - *Value missingness*: If a guideline specifies required tests or symptoms that are not present in the patient record at the current turn, they are encoded as missing. For example, if the CEG requires an $HbA1c$ test but the record does not contain it, this is represented as HbA1c is missing.

  - *Conflict*: A conflict arises when the patient's laboratory test results or symptoms are inconsistent with the diagnostic requirements specified in the retrieved CEG. For example, diabetes diagnosis requires

both an abnormal A1C and an abnormal fasting plasma glucose (FPG). If a patient presents with an abnormal A1C but a normal FPG, this inconsistency is represented as a conflict.

– *Sufficiency*: A patient's symptoms and laboratory test results are considered *sufficient* when they collectively satisfy all diagnostic requirements specified in the retrieved CEG, and the reasoning step reaches its termination node with a diagnosis result.

– *Failure*: We transit to Unresolved when the number of explored CEGs has exceeded maximum limit (which is set to 5 in this work). It represents that the target patient' input does not meet any paths in the retrieved candidate CEGs, and the model falls back to relying on itself for diagnosis.

• $\delta$: the transition function $\delta : \mathcal{S} \times \Sigma \rightarrow \mathcal{S}$ encodes guideline-grounded reasoning rules that map a conversational state and structured inputs to the next state. We define the transition types as follows: (1) transition to PartialResults when the current query satisfies the trigger conditions but has not yet reached a termination (True) state; (2) transition to ConflictResults when patient evidence is inconsistent with CEG requirements, and then re-start the reasoning with the next candidate CEG; (3) transition to ConfirmedDiagnosis when all requirements of the candidate CEG are satisfied, leading to a final diagnosis; (4) when symptom or laboratory test missingness is detected, the DFA will continue the transition to PartialResults during reasoning. (5) transition to Unresolved when diagnosis cannot be made with all the five retrieved CEGs, and generate diagnosis by LLM itself.

### 3.2.2 CONVERSATION GENERATION VIA PATIENT-SPECIFIC DFA

This formalized DFA serves as a powerful tool to align reasoning steps with CEG paths. We prompt the LLM to perform diagnosis using our defined DFA paradigm. The reasoning is an iterative process that includes information seeking, retrieval of relevant CEGs, and state transitions until a final diagnosis is reached. Here are the concrete steps:

**(1) Querying and Retrieving Relevant CEGs.** We first transform patient demographics and symptoms into a retrieval query. We adopt the all-MiniLM-L6-v2 model (Reimers & Gurevych, 2019) to embed the patient's symptoms and each candidate CEG's symptoms. We use cosine similarity to measure their relevance, and select the top-5 candidates with the highest scores. During DFA navigation, these candidates are sequentially selected to guide the reasoning until a termination state is reached.

**(2) DFA Input Generation.** Given a retrieved graph, we generate the input $\Sigma$ for DFA navigation by comparing the patient input with the triggers at the current node. The outcomes are abstracted into structured indicators such as ConditionSatisfied, ValueMissingness, Conflict, and Sufficiency, which serve as inputs driving state transitions.

**(3) Navigating DFA:** Based on $\Sigma$ derived from the patient's input and the retrieved graph, LLMs navigate the DFA along a deterministic path to identify the corresponding state, as defined in Section 3.2.1.

**(4) LLM Response Generation:** This reasoning loop of step 2 and step 3 runs iteratively until a final state is reached, after which we prompt the LLM to make a diagnostic decision based on the reasoning result.

## 4 EXPERIMENTS

We design extensive studies to examine the effectiveness of Autoclinician, covering both retrieval and reasoning aspects. Q1) Can existing LLMs, both general-purpose and medical specialized, perform accurate diagnostic reasoning over EHRs in the absence of clinical guidelines? Q2) Does AutoClinician achieve superior diagnostic accuracy and lab test recommendation performance compared to baselines? Q3) Does AutoClinician properly integrate structural information from clinical guidelines, and what factors influence the effectiveness of guideline incorporation? Q4) Does modeling diagnostic reasoning as a patient-specific DFA provide advantages over CoT prompting or unstructured retrieval? Q5) How does the number of retrieved candidate CEGs affect diagnostic accuracy and the quality of reasoning? Q6) How do clinician evaluate the

quality, faithfulness, and clinical plausibility of the guideline-grounded reasoning produced by AutoClinician?

## 4.1 DATASET AND METRIC DESCRIPTION

To evaluate the performance of LLMs on EHRs, we design datasets based on four principles: (I) *Reality*: Sourced from real-world patient records to reflect authentic clinical workflows, following a pipeline of 'evidence acquisition, result interpretation, and diagnosis confirmation'. (II) *Unexposure*: Ensuring that the corpora have not been previously used by most LLMs. (III) *Open-access*: We use datasets that are publicly available or broadly accessible under standard data-use agreements. (IV) *Predictability*: Focusing on diseases that can be predicted from EHR data. To satisfy these requirements, we adopt two prominent and challenging EHR benchmarks that necessitate multi-turn dialogue reasoning: EHRSHOT (Wornow et al., 2023) and TriNetX (Suarez Arbelaez et al., 2023). Detailed dataset descriptions can be found in Appendix C.1.

We evaluate diagnosis performance using accuracy (ACC) and mean reciprocal rank (MRR), and we also report the laboratory test prediction accuracy (Lab Acc). Details of metrics are shown in Appendix C.2.

## 4.2 BASELINE DESCRIPTION

We compare AutoClinician with comprehensive baselines (1) Vanilla Zero-shot Prompting, including general-purpose LLMs (Qwen3-32B Instruct (Yang et al., 2024a), QwQ-32B (Team, 2025), GPT-4o (Achiam et al., 2023) and medical-specialized models ChatDoctor (Li et al., 2023b), BioMistral (Labrak et al., 2024), Yi-34B (Young et al., 2024) and DiagnosisGPT-34B (Chen et al., 2024)). (2) Few-shot Demonstration with Instructions. We prompt the LLM with a few exemplar dialogues and instructions illustrating a diabetes-related diagnostic reasoning process. (3) Iterative RAG with Query Planning. LLMs will split the diagnostic task into sub-queries through planning, and conduct RAG on guidelines during solving each sub-query.

Table 1: Performance comparison of AutoClinician against baselines on the EHRSHOT dataset, reporting diagnosis accuracy (Acc), mean reciprocal rank (MRR), and laboratory test accuracy (Lab Acc).

| Model | EHRSHOT-Metabolic | | | EHRSHOT-Respiratory | | | EHRSHOT-Circulatory | | |
|---|---|---|---|---|---|---|---|---|---|
| | Acc | MRR | Lab Acc | Acc | MRR | Lab Acc | Acc | MRR | Lab Acc |
| **General-purpose LLMs** | | | | | | | | | |
| Qwen3-32B | 19.73 | 17.81 | 16.61 | 22.31 | 16.15 | 18.08 | 18.47 | 19.40 | 18.73 |
| QwQ-32B | 17.38 | 16.59 | 14.64 | 21.25 | 15.45 | 18.97 | 18.36 | 17.28 | 19.70 |
| GPT-4o | 24.92 | 23.38 | 21.38 | 25.28 | 26.91 | 18.04 | 26.15 | 22.86 | 16.93 |
| **Medical-specialized LLMs** | | | | | | | | | |
| Chatdoctor | 8.54 | 11.26 | 5.09 | 4.62 | 5.73 | 7.09 | 9.93 | 13.65 | 8.17 |
| BioMistral | 9.93 | 12.45 | 8.14 | 7.88 | 8.96 | 8.12 | 9.26 | 12.43 | 8.28 |
| Yi-34B | 16.01 | 16.37 | 16.59 | 17.27 | 18.50 | 16.58 | 17.09 | 16.31 | 16.81 |
| DiagnosisGPT-34B | 17.02 | 18.07 | 18.27 | 19.95 | 18.35 | 18.51 | 18.17 | 18.58 | 18.01 |
| **Few-shot Demo. w/ Instruction** | | | | | | | | | |
| Qwen3-32B | 18.77 | 15.89 | 15.94 | 20.49 | 14.83 | 16.96 | 17.07 | 18.99 | 16.67 |
| QwQ-32B | 15.38 | 14.77 | 20.93 | 19.94 | 13.92 | 17.14 | 15.16 | 13.52 | 20.07 |
| **Enhancing Reasoning with iterative RAG** | | | | | | | | | |
| RAG-Qwen3-32B | | | | | | | | | |
| w/ Query Planning | 21.30 | 22.81 | 20.10 | 24.31 | 25.15 | 21.08 | 24.47 | 22.40 | 20.73 |
| w/ **Autoclinician** | 26.54 | 26.92 | 23.37 | 29.55 | 31.09 | 27.32 | 30.68 | 29.14 | 24.98 |
| RAG-QwQ-32B | | | | | | | | | |
| w/ Query Planning | 20.38 | 21.59 | 19.64 | 22.25 | 23.45 | 21.97 | 23.36 | 22.28 | 21.70 |
| w/ **Autoclinician** | 26.72 | 27.48 | 24.59 | 27.08 | 29.71 | 25.34 | 27.62 | 26.87 | 27.15 |

## 4.3 MAIN RESULTS (Q1 & Q2)

For space limitation, experimental results are presented in Table 1 and Table 5 in Appendix C.3. We can make the following observations: (1) **Vanilla zero-shot prompting yields limited diagnostic performance on EHRs reasoning.** Both general-purpose and medical-specialized LLMs achieve low accuracy and MRR,

revealing that they lack deep understanding of EHR data and have little domain knowledge to support diagnostic reasoning. (2) **Few-shot prompting with instructions does not improve diagnostic reasoning ability and can even degrade performance.** This result reflects both the limited reasoning ability of LLMs in complex diagnostic workflows and the insufficiency of few examples in providing generalizable diagnosis knowledge. (3) **Incorporating clinical guidelines substantially boosts performance.** RAG with query planning methods outperform other baselines with a clear margin, demonstrating that structured clinical evidence is essential for accurate diagnostic reasoning. Furthermore, AutoClinician demonstrates remarkable performance gain over others, underscoring the advantage of our proposed structured evidence integration.

### 4.4 DETAILED ANALYSIS

**Importance of CEGs (Q3).** We investigate the contribution of guideline compression through a series of ablations, with results summarized in Figure 2 and Figure 3. Specifically, we compare with: (1) AutoClinician w/o Self-refinement, which removes the self-consistency refinement applied during CEG construction; and (2) AutoClinician w/o CEGs, which appends the original guideline text directly to input prompts.

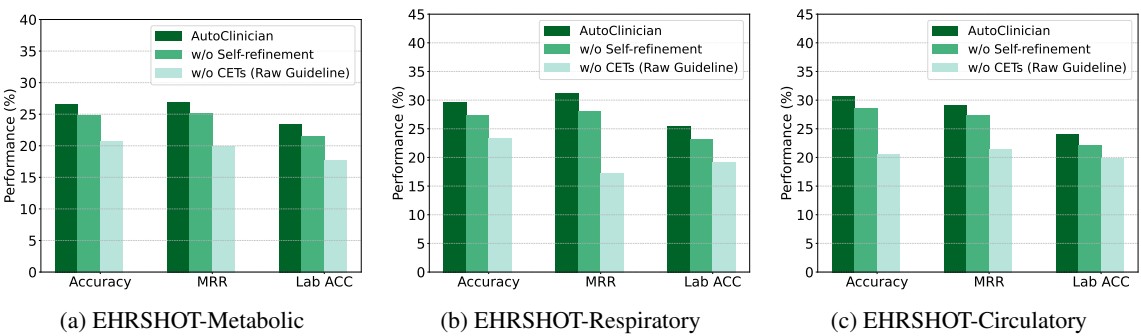

Figure 2: Ablation study of CEGs on the EHRSHOT dataset.

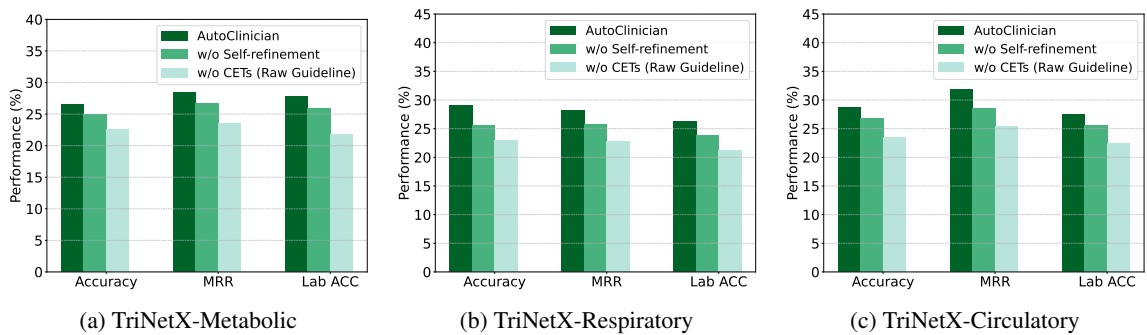

Figure 3: Ablation study of CEGs on the TriNetX dataset.

AutoClinician consistently achieves the highest performance, highlighting the critical role of leveraging structured knowledge from clinical guidelines for diagnostic reasoning. Eliminating self-refinement produces a moderate decline, indicating that refinement improves the qualities of CEGs. Removing CEGs performs better than zero-shot prompting but remain substantially worse than AutoClinician. This indicates that compressing guideline text help LLMs to focus on essential logical dependencies, thereby improving its capacity for multi-step diagnostic reasoning.

**Importance of DFA (Q4).** We conduct ablation studies to assess the effect of modeling diagnostic reasoning as a DFA within AutoClinician, as shown in Table 2 and Table 3. Specifically, we compare against two variants: (1) AutoClinician w/o DFA, which removes the state tracker and directly augments the prompts with the retrieved CEG; (2) CoT Prompting, which replaces structured DFA with manually written reasoning cues.

Table 2: Ablation study of DFA on the EHRSHOT dataset.

| Reasoning Method | EHRSHOT-Metabolic | | | EHRSHOT-Respiratory | | | EHRSHOT-Circulatory | | |
|---|---|---|---|---|---|---|---|---|---|
| | Acc | MRR | Lab ACC | Acc | MRR | Lab ACC | Acc | MRR | Lab ACC |
| AutoClinician | 26.54 | 26.92 | 23.37 | 29.55 | 31.09 | 27.32 | 30.68 | 29.14 | 24.98 |
| w/o DFA | 20.83 | 20.30 | 18.55 | 24.73 | 24.53 | 22.76 | 25.28 | 23.05 | 19.74 |
| CoT | 21.12 | 22.62 | 20.28 | 25.11 | 25.86 | 23.32 | 24.96 | 24.58 | 22.75 |

Table 3: Ablation Study of DFA on the TriNetX dataset.

| Reasoning Method | TriNetX-Metabolic | | | TriNetX-Respiratory | | | TriNetX-Circulatory | | |
|---|---|---|---|---|---|---|---|---|---|
| | Accuracy | MRR | Lab Acc | Accuracy | MRR | Lab Acc | Accuracy | MRR | Lab Acc |
| **AutoClinician** | 26.50 | 28.50 | 27.80 | 29.00 | 28.20 | 26.33 | 28.70 | 31.80 | 27.40 |
| **w/o DFA** | 22.76 | 23.21 | 22.45 | 23.83 | 22.60 | 20.81 | 23.91 | 24.17 | 23.22 |
| **CoT** | 22.30 | 21.88 | 20.01 | 24.47 | 23.65 | 23.22 | 23.45 | 25.41 | 24.90 |

AutoClinician achieves the highest performance across all datasets, highlighting the importance of maintaining and updating patient-specific reasoning states derived from both the EHR and CEGs. Accurate reasoning requires conducting sufficient laboratory tests, interpreting value thresholds, and integrating multiple clinical findings for joint evaluation, leading to AutoClinician's improved performance over CoT.

**Effect of the Number of CEG Candidates (Q5).** We further analyze the model's sensitivity to the number of candidate CEGs. Experiments are conducted on the EHRSHOT dataset with Qwen3-32B as the backbone, with results summarized in Table 4. It can be observed that increasing the candidate number leads to higher accuracy score. We leave the exploration of more advanced retrieval mechanisms to future work.

Table 4: Accuracy of AutoClinician with different number of candidate CEGs across three EHRSHOT subsets.

| Model | Metabolic | | | | Respiratory | | | | Circulatory | | | |
|---|---|---|---|---|---|---|---|---|---|---|---|---|
| | Acc@1 | Acc@3 | Acc@5 | Acc@10 | Acc@1 | Acc@3 | Acc@5 | Acc@10 | Acc@1 | Acc@3 | Acc@5 | Acc@10 |
| AutoClinician | 19.47 | 23.31 | 26.54 | 27.95 | 21.12 | 26.29 | 29.55 | 31.61 | 18.86 | 22.15 | 30.68 | 30.76 |

**Human Evaluation and Case Study (Q6).** We perform a case study with the collaboration of clinicians to review approximately 30 generated dialogues and evaluate the guideline-grounded reasoning from multiple perspectives, including guideline adherence, laboratory test interpretation, conflict and ambiguity resolution, and overall clinical plausibility. Detailed evaluation results are provided in the Appendix C.4.3.

## 5 CONCLUSIONS AND FUTURE WORKS

In this work, we introduced AutoClinician, a training-free framework that integrates structured clinical guidelines into EHR-based diagnostic conversations. We first compress clinical guidelines into logical evidence graphs and propose a self-consistency srategy to evaluate and refine the generated structures. Diagnostic reasoning is further modeled as a Deterministic Finite Automaton (DFA) that tracks patient-specific diagnostic states using sequential EHR-based inputs and guideline-derived graphs. Experiments demonstrate that AutoClinician consistently outperforms both general-purpose and clinical-specialized LLMs. Future work will extend AutoClinician beyond EHR-only prediction by incorporating multimodal input and aligning guidelines with richer diagnostic inputs. Another direction is to develop AutoClinician into a web-based agent that dynamically leverages public medical resources to support evidence-grounded patient self-interpretation and physician decision-making.

## ETHICAL STATEMENT

All data used in this study are fully de-identified. The datasets are available to researchers for academic purposes, and access to certain datasets and documents requires institutional membership and appropriate approvals. No personally identifiable information (PII) or sensitive clinical records were used. This study is intended to model clinical reasoning processes and to use LLMs for response generation, without any attempt to identify, profile, or make inferences about real individuals.

## REPRODUCIBILITY STATEMENT

We are committed to ensuring the reproducibility of our work. All datasets used in this study are either publicly available or accessible to academic researchers under appropriate agreements. We provide the full implementation, including preprocessing scripts and modeling pipelines, via an anonymous link in this paper. The prompts used in our experiments as well as detailed implementation descriptions are included in the Appendix. These resources enable independent researchers to reproduce our results and extend our work.

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

APPENDIX

## A    THE USE OF LARGE LANGUAGE MODELS

Large Language Models (LLMs) were employed in this project in a limited, assistive capacity. For manuscript preparation, all sections were drafted by the authors, with LLMs occasionally used to improve grammar, enhance clarity, and refine academic tone. During the implementation phase, LLMs served as coding assistants, providing code completions, debugging suggestions, and formatting support. However, all final code, experimental design, analyses, and validations were developed and verified solely by the authors. Importantly, LLMs were not used for generating research ideas, designing experiments, or conducting the literature review. All conceptual contributions, methodological innovations, and scientific insights originated exclusively from the authors.

## B    IMPLEMENTATION

### B.1    IMPLEMENTATION DETAILS

In our experiments, we first extract diagnosis-relevant text from clinical guidelines and then summarize this content into Clinical Evidence Graphs (CEGs). Each CEG consists of a set of symptoms serving as query targets and a graph that represents the diagnostic logic derived from the guideline. We utilize the all-MiniLM-L6-v2 model (Reimers & Gurevych, 2019) as the embedding model to index both the CEGs and input queries. At test time, each input query is encoded and compared against all embeddings in the corpus, and the top-5 nearest neighbors are retrieved for inference.

For HTML-based guidelines[3], we use the Python library `lxml` to traverse the DOM structure and extract section headers and corresponding content blocks. Tags such as `<h2>`, `<p>`, and `<li>` are used to identify semantic structure. For PDF-based guidelines, we convert documents to structured HTML using the Python library PyMuPDF, preserving layout features including heading fonts, indentation, and section boundaries. We then apply regular expression and keyword-based filters to identify diagnosis-relevant sections. Candidate sections are selected based on the presence of headings or sentence patterns containing terms such as "diagnosis," "screening," "classification," "introduction," or "criteria." The extracted content is treated as diagnosis-related context and serves as input for CEGs generation.

For Clinical Evidence Graph (CEG) generation, we utilize GPT-4o for more accurate guideline parsing. We first extract diagnosis-related sections of clinical guidelines with the prompt in Figure 4, and the extracted text is then passed to the CEG generation prompt (see Figure 5). The model is instructed to output a structured JSON object consisting of verbatim symptoms as query key and step-wise diagnostic reasoning represented as graph nodes. Each node links context, trigger, and state, with deterministic transitions ensuring that every non-terminal state matches the context of a subsequent node. In our preliminary experiments, we identified four types of potential errors in CEGs. To address these, we generated pseudo-patient samples directly from the diagnosis-related content (as shown in Figure 6), and then used these samples to refine the CEGs (as shown in Figure 7). We exclude clinical tests with a frequency of less than 10 across the entire dataset. Additionally, we filter out tests for which no test results are available for each patient.

In our experiments, LLMs are instructed to output answers in descending order of confidence, forming a ranked list. We adopt the all-MiniLM-L6-v2 model (Reimers & Gurevych, 2019) to embed both the LLMs' responses and the ground-truth responses for diagnoses or laboratory tests. When the base model is QwQ-32B, we additionally employ Qwen3-14B to extract and summarize reasoning results, which are then compared against the ground truth for evaluation.

---

[3] https://diabetesjournals.org/care/article/48/Supplement_1/S27/157566/2-Diagnosis-and-Classification-of-Diabetes

To enable semantic processing of structured EHR data, we mapped coded medical concepts to their textual descriptions. Specifically, we downloaded three standardized clinical vocabularies: (1) International Classification of Diseases (ICD) codes from the World Health Organization (WHO, https://www.who.int/standards/classifications/classification-of-diseases), (2) the SNOMED CT International Release (20250801) from the SNOMED International Member Licensing and Distribution Service (MLDS, https://mlds.ihtsdotools.org), and (3) RxNorm (2025AA) from the U.S. National Library of Medicine (NLM, https://www.nlm.nih.gov/research/umls/rxnorm). Each resource provides machine-readable mappings from coded identifiers to human-readable terms, which we use to convert diagnosis, procedure, and medication codes into natural language representations. For example, the ICD-10 code E11.9 corresponds to the description "Type 2 diabetes mellitus without complications," the SNOMED CT concept 44054006 maps to "Diabetes mellitus type 2 (disorder)," and the RxNorm code 860975 represents "Metformin 500 MG Oral Tablet." These mappings allow EHR data to be directly integrated into text.

## B.2 Example Prompt for Extracting Text from Clinical Guidelines

---

**Prompt for Extracting Diagnosis-related Text from Clinical Guidelines**

You are a clinical information extractor. You read guideline pages and extract verbatim diagnosis-relevant content.

Input: `<An Guideline file>` describing `<Guideline Title>`.

Task: From the guideline file, localize and extract only the sections that directly support diagnosis prediction, including (but not limited to):
1. Symptoms/signs/demographics explicitly mentioned (verbatim phrases only)
2. Diagnostic lab tests (test names, specimen, timing, test procedures) 3. Cut-points / thresholds and normal/abnormal ranges (all units included)
4. Conditions/contraindications/limitations where a test should not be used or is unreliable
5. Confirmation rules (e.g., repeat testing, discordant results handling)
6. Risk-based screening criteria only if they contain thresholds or explicit test instructions relevant to diagnosis

Inclusions and Exclusions:
1. Include only content that is directly diagnosable signals (symptoms/signs), test definitions, numeric thresholds/ranges, confirmation rules, and test limitations.
2. Exclude epidemiology, pathophysiology discussions, economic or implementation commentary, treatment/management recommendations, and non-diagnostic background unless it contains explicit diagnostic thresholds or rules.

Requirements: Follow the original document order. For each extracted block, add the original section heading (verbatim) as a json header before the content. Quality and Formatting requirements (must-follow):
1. Output one json file only.
2. Use original text verbatim; no rephrasing.
3. For tables, use Markdown tables; do not insert long sentences into tables-only the cells as in source.
4. Preserve units, symbols (e.g., >,=,<), and footnote markers.
5. If a referenced table/figure is outside the text but the table is present, include it. If a table is an image, write a short line: Table image present.
6. If a section references another table (e.g., "Table 2.2"), and that table exists in the guideline, include it next.
7. If any segment cannot be confidently localized, add a one line placeholder Section not found in guideline and continue.

---

Figure 4: Prompt for Extracting Text from Clinical Guidelines. This prompt instructs a LLM to transform clinical guideline into a structured json file that preserves verbatim diagnosis-relevant evidence (symptoms, tests, thresholds, confirmation rules, and limitations) while excluding all non-diagnostic material.

Example prompt is shown in Figure 4.

## B.3 EXAMPLE PROMPT FOR CLINICAL EVIDENCE GRAPH GENERATION

---

**Example Prompt for Generating CEGs**

You are an clinical graduates trained to undertand clinical guideline texts and extract structured diagnostic logic. Given a clinical guideline text focused on diagnosis, your task is to output a JSON object that contains:

1. Symptom: A list of clinical symptoms, patient demographics, or signs explicitly mentioned in the text that are associated with the diagnosis. Extract what is written and do not infer or paraphrase. Use the exact wording from the text.

2. Graphs: A sequence of diagnostic reasoning steps represented in the following format:
  - ID: Denotes the index of current node in the graph.
  - Context': The current diagnostic state or clinical situation. This links to the '"state"' of the previous step.
  - Trigger: Specifies the clinical condition or threshold defined by guidelines. A threshold, condition, lab result, or observed symptom that triggers diagnosis progression.
  - State: Specifies the guideline-prescribed clinical action taken when the Trigger is activated. It reflects how to act upon the observed condition based on clinical guidelines. For example, 'State = Suspect diabetes' when the patient's laboratory tests satisfy the defined Trigger.
  - Termination: true or false. If true, this node is a leaf (final diagnosis or conclusion). If false, the engine will continue by locating the next node where context equals to this node state.
For every node with "terminal": false, the value of "state" MUST exactly match the "context" of a subsequent node. Do NOT invent text.

Instructions:
  - Use only information that is explicitly stated in the input text. Maintain the chain of reasoning by linking each step 'context' to the 'state'' of the prior step.
  - Output the result strictly in JSON format.
  - Do not include any commentary, explanation, or extra text, only the JSON object.

Input (diagnosis-related sections): <INSERT TEXT HERE>
Output JSON format only. Output format:
  "Symptom": ["<verbatim symptom/sign/demographic>", "..."],
  "Graphs": [ "Context": "<where we are now>", "Trigger": ["<verbatim trigger 1>", "<verbatim trigger 2>"], "State": "<resulting status>", "Terminal": False , "Context": "<must equal previous state if previous Terminal=false>", "Trigger": ["<verbatim trigger>"], "State": "<final diagnosis or next status>", "Terminal": True ]

---

Figure 5: Example Prompt for Generating CEGs. This prompt instructs the model to convert diagnosis-related guideline text into a structured JSON-based Clinical Evidence Graph, capturing verbatim symptoms and stepwise diagnostic reasoning logic.

Example prompt is shown in Figure 5.

## B.4 EXAMPLE PROMPT FOR CLINICAL EVIDENCE GRAPH REFINEMENT

Prompt for pseudo-patient example generation is shown in Figure 6. If the alignment score between pseudo-patient examples and CEGs falls below 0.8, we prompt LLM with failed pseudo-patient examples to refine the initial CEGs. Example prompt for CEGs refinement is shown in Figure 7.

---

**Example Prompt for Generating Pseudo-patient Examples**

You are a clinical reasoning expert. You are given a diagnosis-related clinical guideline passage `INPUT DIAGBISTIC RELATED SECTIONS`. Your task is to generate synthetic patient cases to test the robustness of a diagnostic logic graph automatically compressed from that guideline.

Follow these instructions:
    1. **Read the guideline** carefully and identify key diagnostic criteria: symptoms, lab thresholds, age/BMI ranges, and test sequences.
    2. **Generate pseudo-patient example samples** that intentionally challenge the logic tree understanding. Each sample should fall into one of the following categories:
      - Incomplete Evidence: A required symptom, test, or condition is missing.
      - Threshold Misalignment: The input includes a value just above or below a diagnostic threshold.
      - Sequential Logic Error: The order of diagnosis steps is violated (e.g., insulin started before testing).
      - Contradictory Evidence Injection: The patient presents with mixed evidence that makes diagnosis ambiguous.
    3. For each sample, return the following fields in JSON format:
- Patient Input: Include age, BMI, symptoms, lab results, relevant family or autoimmune history.
- Type: One of: '"Incomplete Evidence"', '"Threshold Misalignment"', '"Sequential Logic Error"', or '"Contradictory Evidence Injection"'.
- Expected Output: What the correct diagnosis should be according to the guideline.
- Guideline Evidence: A quoted or paraphrased sentence from the guideline that justifies the expected output.

Please generate 15 samples for each of the four types (total around 50 examples), and ensure all reasoning strictly follows the logic stated in the guideline.

---

Figure 6: Prompt for Generating Pseudo-patient Examples. This prompt is used to generate structured pseudo-patient examples that deliberately challenge diagnostic logic graphs, ensuring robustness against incomplete evidence, borderline thresholds, sequential errors, and contradictory clinical findings.

### B.5 PROMPTS FOR DIALOGUE GENERATION USING DFA

Example Prompt for using DFA as a dialogue tracker can be found in Figure 8.

### B.6 COT PROMPTS

Example prompt for CoT used in baselines can be found in Figure 9.

## C ADDITIONAL EXPERIMENTS

### C.1 DATASET DESCRIPTION

**EHRSHOT** is a de-identified, public large-scale benchmark of electronic health records from 6,739 adult patients at Stanford Medicine (Wornow et al., 2023) . It contains 41.6 million structured clinical events across 921,499 encounters, spanning demographics, diagnoses, procedures, laboratory results, and medications.

**TriNetX Dataset**[4] is a global health research network providing access to large-scale de-identified patient EHR data (Suarez Arbelaez et al., 2023). TriNetX includes records from 33,105 de-identified patients, spanning data from 1982 to 2023, collected across more than 100 community hospitals and 500 outpatient clinics.

---

[4]https://trinetx.com/.

---

**Prompt for Refining CEGs using Failed Pseudo-patient Examples.**

You are given a Clinical Evidence Graph (CEG) represented as nodes with attributes <ID, Context, Trigger, State, Termination>. You are also provided with a set of pseudo-patient examples, each synthesized directly from the guideline text and containing verbatim guideline conditions (e.g., laboratory thresholds, diagnostic criteria, symptom descriptions).

Here is the CEGs `<CEGs input>` and pseudo-example `<pseudo-examples>` with the type of `<error type>`.

Instructions:
For each pseudo-example, follow the reasoning path in the CEG by matching the Trigger → State → Context transitions. If a pseudo-example cannot be mapped consistently, identify the misaligned nodes.

Refine the CEG by:
Pruning inconsistent or redundant nodes.
Adjusting node attributes only using conditions and wording explicitly stated in the guideline text.
Ensuring that all pseudo-examples can be faithfully explained by the refined CEG.

Output format:
Return the refined CEG in the same 5-tuple format <ID, Context, Trigger, State, Termination>. Additionally, include a concise note summarizing which nodes were modified or pruned, and cite the exact guideline phrase used for the refinement.

Figure 7: Prompt for Refining CEGs using failed pseudo-patient examples.

## C.2 EVALUATION METRICS

In our experiments, LLMs are instructed to output answers in descending order of their confidence, forming a prioritized ranked list. The Mean Reciprocal Rank (MRR) (Jordan & Mitchell, 2015) is a metric used to evaluate agents that return a ranked list of answers to queries, focusing on the position of the first relevant answer. A higher MRR indicates that relevant items tend to appear higher in the ranked list of results. It is defined as the average of the reciprocal ranks of the first relevant answer for a set of queries.

## C.3 MAIN RESULTS

Our main results on TrinexT dataset is shown in Table 5.

## C.4 DETAILED ANALYSIS

### C.4.1 EXAMPLES OF THE GENERATED PSEUDO-PATIENT SAMPLES

We generate four types of pseudo-patient examples to automatically validate and refine CEGs: (1) ***Evidence Modification***. For example, we randomly remove a symptom or a required lab test. Examples are shown in Figure 10. (2) ***Boundary Threshold Perturbation***. For example, given a guideline threshold of 100 mg/dL, a perturbed case at 98 mg/dL should be rejected by the tree to ensure threshold adherence. Examples are shown in Figure 11. (3) ***Step Reordering*** that reverses the order of diagnostic steps. Examples are shown in Figure 12. (4) ***Contradictory Evidence Injection*** that introduces inconsistency between test results. Examples are shown in Figure 13.

### C.4.2 HUMAN EVALUATION ON CEGS AND CASE STUDY

**Example Prompt for Dialogue Reasoning using DFA**

You operate a Deterministic Finite Automaton (DFA) defined as $(S, \Sigma, \delta, s_0, F)$.

$S$: a finite set of conversational states with start state $s_0$ and terminal states $F$.

$\Sigma$: input categories derived from CEG-guided reasoning over the patient's dialogue (based on Retrieved CEG and patient Input).

$\delta$: a deterministic transition function (provided in DFA specification).

**Requirements:**

1) Read `<Retrieved CEG>` and the user's dialogue to determine which triggers are satisfied, which values are missing, and whether results are discordant.

2) From those conclusions, derive $\Sigma$ at each step (choose one of: `ConditionSatisfied`, `Missingness`, `Conflict`, `Sufficiency`, `Failure`).

3) Apply $\delta$ deterministically to move from the current state in $S$ to the next, repeating step by step for up to `MAX_STEPS=15`, or stop early at a terminal state.

4) Perform all intermediate reasoning internally; **DO NOT** reveal intermediate steps.

**DFA specification:**

$$S = \{\text{Start, PartialResults, ConflictResults, ConfirmedDiagnosis,}$$
$$\text{Unresolved, AnotherCandidate}\}$$
$$\Sigma = \{\text{ConditionSatisfied, Missingness, Conflict, Sufficiency, Failure}\}$$
$$\delta(s, \text{Missingness}) \rightarrow \text{PartialResults}$$
$$\delta(s, \text{ConditionSatisfied}) \rightarrow \text{PartialResults (until a terminal True node is reached)}$$
$$\delta(s, \text{Conflict}) \rightarrow \text{ConflictResults}$$
$$\delta(s, \text{Sufficiency}) \rightarrow \text{ConfirmedDiagnosis}$$
$$\delta(s, \text{Failure}) \rightarrow \text{AnotherCandidate}$$
$$\text{Terminal states} = \{\text{ConfirmedDiagnosis, Unresolved}\}$$

**Retrieved CEG:**

`<Input Retrieved CEGs>`
FORMAT: Return ONE sentence only.

`<Current Patient Input>`

**Output Requirement:**

Output **EXACTLY ONE** sentence in plain English, ending with a period. The sentence MUST include:

(a) a concise, guideline-based explanation of the patient's current status (from CEG + inputs), and

(b) the current state and the most appropriate next step implied by your DFA traversal (e.g., which test is missing, whether results conflict, or that diagnosis is confirmed).

No bullet points, no lists, no JSON, no headings, no code fences, no extra lines and ONE sentence only.

Figure 8: Example prompt used for dialogue reasoning with a DFA, specifying the state set, input categories, and transition rules, along with requirements for processing retrieved CEGs and patient inputs to generate a guideline-grounded sentence summarizing the patient's status and next step.

**CoT Prompts.**

You are a clinical expert assistant. Given a patient's EHR, reason step by step to reach a diagnosis. First, examine the symptoms and lab results. Then, check if diagnostic thresholds are met. Finally, determine the disease subtype and provide a diagnosis based on medical guidelines.
Patient EHR: <insert input>
Step-by-step reasoning:
1. What symptoms does the patient report? Are they consistent with any common disease?
2. What are the clinical semantic meaning of lab test results, e.g., abnormal or normal? Are any lab test results above diagnostic thresholds?
3. Based on symptoms and lab data, can we confirm a diagnosis?
4. What is the likely disease subtype based on age, BMI, and symptom onset?
5. What is the final diagnosis?
Answer:

Figure 9: Example of Cot Prompt used in Baselines.

**Examples for Incomplete Evidence.**

Example 1:
    "Patient Input": "age": 29, "BMI": 27.2, "symptoms": [], "A1C": 6.6 %, "pregnant": true, "trimester": 2,
    "Type": "Incomplete Evidence.",
    "Expected Output": "Not diagnostic; Do not use A1C to diagnose in pregnancy; apply plasma glucose criteria.",
    "Guideline Evidence": "In conditions… pregnancy (second and third trimesters…) only plasma blood glucose criteria should be used to diagnose diabetes."
Example 2:
    "Patient Input": "age": 46, "BMI": 31.4, "symptoms": ["fatigue"], "FPG": 128 mg dL, "fasting_hours": 5,
"A1C": 6.3 %,
    "Type": "Incomplete Evidence.",
    "Expected Output": "Not diagnostic; fasting status invalid (<8 h). Repeat FPG with $\geq$ 8 h fast or use another test.", "Guideline Evidence": "FPG $\geq$ 126 mg/dL… Fasting is defined as no caloric intake for at least 8 h."
Example 3:
    "Patient Input": "age": 41, "BMI": 28.0, "symptoms": [], "two hour OGTT": 202 mg/dL, "glucose": 75 g,
"carb prep days": null,
    "Type": "Incomplete Evidence.",
    "Expected Output": "Not diagnostic; Potentially invalid OGTT; ensure $\geq$ 150 g/day carbohydrate for 3 days prior, then retest.",
    "Guideline Evidence": "Adequate carbohydrate intake (at least 150 g/day) should be assured for 3 days prior to oral glucose tolerance testing…"

Figure 10: Pseudo-patient Examples for Incomplete Evidence.

---

**Pseudo-patient Examples for Threshold Misalignment.**

Example 1:
"Patient Input": "age": 60, "BMI": 30.1, "symptoms": [], "FPG": 125 mg/dL, "fasting": 9 hours,
"Type": "Threshold Misalignment",
"Expected Output": "Does not meet diabetes threshold; near threshold—repeat or perform OGTT; consider retesting in 3–6 months if near margins.",
"Guideline Evidence": "FPG $\geq$ 126 mg/dL" and "If patients have test results near the margins... repeat the test in 3–6 months."

Example 2:
"Patient Input": "age": 39, "BMI": 27.6, "symptoms": [], "A1C": 6.4 %,
"Type": "Threshold Misalignment",
"Expected Output": "Below diagnostic threshold; not diabetes; repeat/alternative testing if near margins.",
"Guideline Evidence": "A1C $\geq$ 6.5%..." and "If... near the margins... repeat... in 3–6 months."

Example 3:
"Patient Input": "age": 50, "BMI": 31.8, "symptoms": ["polyuria", "polydipsia"], "random glucose": 199 mg/dL
"Type": "Threshold Misalignment",
"Expected Output": "Not diagnostic; random value is below 200 mg/dL even with symptoms; perform/confirm with FPG, 2-h OGTT, or A1C.",    "Guideline Evidence": "In a patient with classic symptoms... random plasma glucose $\geq$ 200 mg/dL."

Figure 11: Pseudo-patient Examples for Threshold Misalignment.

---

**Pseudo-patient Examples for Logic Error.**

<Example 1>:
"Patient Input": "age": 56, "BMI": 29.7, "symptoms": [], "random glucose": 205 mg/dL, "diagnosis declared before confirmation": true,
"Type": "Sequential Logic Error",
"Expected Output": "Do not diagnose yet; without classic symptoms, obtain confirmatory abnormal testing.",
"Guideline Evidence": ""In the absence of unequivocal hyperglycemia, diagnosis requires two abnormal test results...""

<Example 2>:
"Patient Input": "age": 64, "BMI": 27.9, "symptoms": [], "A1C %": 6.6, "FPG": 99 mg/dL, "repeat_timing": "delayed months",
"Type": "Sequential Logic Error",
"Expected Output": "Repeat the above-threshold test without delay; do not postpone confirmation for months.",
"Guideline Evidence": ""It is recommended that the second test... be performed without delay.""

<Example 3>:
"Patient Input": "age": 30, "BMI": 28.3, "symptoms": [], "pregnant": true, "A1C": 6.7%, "plasma tests": "not ordered",
"Type": "Sequential Logic Error",
"Expected Output": "Incorrect sequence; in pregnancy, use plasma glucose criteria (order glucose testing) rather than A1C.",
"Guideline Evidence": ""In conditions... pregnancy... only plasma blood glucose criteria should be used to diagnose diabetes.""

Figure 12: Pseudo-patient Examples for Logic Error.

Table 5: Performance comparison of AutoClinician against baselines on the TrinetX dataset, reporting diagnosis accuracy (Acc), mean reciprocal rank (MRR), and laboratory test accuracy (Lab Acc).

| Model | TriNetX-Metabolic | | | TriNetX-Respiratory | | | TriNetX-Circulatory | | |
|---|---|---|---|---|---|---|---|---|---|
| | Accuracy | MRR | Lab ACC | Accuracy | MRR | Lab ACC | Accuracy | MRR | Lab ACC |
| Vanilla Zero-shot Prompting (*Baselines w/o Retrieval*) | | | | | | | | | |
| **General-purpose LLMs** | | | | | | | | | |
| Qwen3-32B | 19.63 | 18.34 | 21.34 | 20.21 | 18.17 | 17.83 | 20.59 | 19.86 | 21.16 |
| QwQ-32B | 23.96 | 20.45 | 23.47 | 18.56 | 15.41 | 14.35 | 17.34 | 20.16 | 19.21 |
| GPT-4o | 26.28 | 21.56 | 20.01 | 24.83 | 21.52 | 23.73 | 25.84 | 23.96 | 24.82 |
| **Medical-specialized LLMs** | | | | | | | | | |
| Chatdoctor | 6.87 | 18.92 | 1.77 | 2.89 | 9.44 | 0.95 | 16.24 | 22.08 | 1.18 |
| BioMistral | 9.37 | 16.62 | 2.33 | 2.67 | 3.98 | 1.70 | 4.44 | 6.56 | 5.79 |
| Yi-4B | 14.06 | 13.28 | 11.21 | 18.89 | 14.67 | 16.79 | 12.22 | 21.90 | 11.29 |
| DiagnosisGPT-34B | 16.22 | 21.38 | 2.90 | 17.33 | 24.32 | 1.37 | 18.22 | 17.26 | 9.91 |
| **Few-shot Demo. w/ Instruction** | | | | | | | | | |
| Qwen3-32B | 20.57 | 16.82 | 19.75 | 23.45 | 17.23 | 18.44 | 21.59 | 18.06 | 20.36 |
| QwQ-32B | 21.57 | 18.91 | 13.52 | 19.89 | 16.06 | 14.81 | 22.51 | 23.31 | 18.20 |
| **Enhancing Reasoning with iterative RAG** | | | | | | | | | |
| RAG-Qwen3-32B | | | | | | | | | |
| w/ Query Planning | 22.85 | 23.81 | 22.10 | 24.31 | 23.15 | 22.08 | 22.47 | 26.40 | 24.73 |
| w/ **Autoclinician** | 26.50 | 28.50 | 27.80 | 29.00 | 28.20 | 26.33 | 28.70 | 31.80 | 27.40 |
| RAG-QwQ-32B | | | | | | | | | |
| w/ Query Planning | 25.37 | 26.59 | 18.64 | 23.25 | 25.45 | 24.97 | 23.36 | 25.28 | 25.33 |
| w/ **Autoclinician** | 31.50 | 31.20 | 26.37 | 27.10 | 26.41 | 29.50 | 32.79 | 33.02 | 30.28 |

---

**Pseudo-patient Examples for Contradictory Evidence.**

<Example 1>:
"Patient Input": "age": 28, "BMI": 23.8, "symptoms": [], "A1C": 6.9 %, "FPG": 92 mg/dL, "sickle cell": true,
"Type": "Contradictory Evidence Injection",
"Expected Output": "Do not use A1C in sickle cell disease; diagnose only via plasma glucose criteria.",
"Guideline Evidence": ""…hemoglobinopathies… only plasma blood glucose criteria should be used…"
and "Marked discordance… should raise the possibility of A1C assay interference.""

<Example 2>:
"Patient Input": "age": 52, "BMI": 31.2, "symptoms": [], "random glucose": 210 mg/dL, "A1C": 6.4%,
"Type": "Contradictory Evidence Injection",
"Expected Output": "Not diagnostic without classic symptoms; proceed with FPG/OGTT/A1C confirmation.",
"Guideline Evidence": ""In a patient with classic symptoms… random plasma glucose $\geq$ 200 mg/dL.""

<Example 3>:
"Patient Input": "age": 70, "BMI": 24.6, "symptoms": [], "hemodialysis": true, "A1C ": 6.8%, "FPG ": 102 mg/dL,
"Type": "Contradictory Evidence Injection",
"Expected Output": "Do not diagnose by A1C in hemodialysis; require plasma glucose criteria.",
"Guideline Evidence": ""…hemodialysis… only plasma blood glucose criteria should be used…""

Figure 13: Pseudo-patient Examples for Contradictory Evidence.

We collaborated with clinicians to review metabolic-related CEGs and selected one diabetes-related case as a representative example. Human experts then evaluated the CEGs in Table 14 along three dimensions: guideline faithfulness, Logic coherence, and clinical utility.

**One example of Generated Diabete-related CEGs**

```
{
  "Symptom": {
    "classic symptoms of hyperglycemia",
    "hyperglycemic crisis",
    "pregnancy (second and third trimesters and the postpartum period)",
    "hemoglobinopathies including sickle cell disease",
    "glucose−6−phosphate dehydrogenase deficiency",
    "HIV",
    "hemodialysis",
    "recent blood loss or transfusion",
    "erythropoietin therapy",
    "symptomatic patients" },

  "Graphs": {
    "Nodes": {
      {
       "ID": "S1",
       "Context": "Diagnostic Tests for Diabetes",
       "Trigger": "Diabetes may be diagnosed based on plasma glucose criteria, either the fasting plasma
         glucose (FPG) value or the 2−h plasma glucose (2−h PG) value during a 75−g oral glucose tolerance test
          (OGTT) or A1C criteria",
       "State": "Criteria for the diagnosis of diabetes",
       "Terminal": false
      },
      {
       "ID": "S2",
       "Context": "Criteria for the diagnosis of diabetes",
       "Trigger": "FPG  126 mg/dL (7.0 mmol/L). Fasting is defined as no caloric intake for at least 8 h.",
       "State": "Confirming the Diagnosis",
       "Terminal": false
      },
          ...

      {
       "ID": "S14",
       "Context": "Multiple confirmed islet autoantibodies is a risk factor for clinical diabetes",
       "Trigger": "When multiple islet autoantibodies are identified, referral to a specialized center for further
         evaluation and/or consideration of a clinical trial or approved therapy to potentially delay development
         of clinical diabetes should be considered.",
       "State": "High risk of type 1 diabetes (pre−clinical stage)",
       "Terminal": true
      }
    }}}
```

Figure 14: This result presents a JSON output of the generated CEGs for diabetes diagnosis, including a list of relevant symptoms used for query embedding alongside a sequence of numbered logic nodes (S1–S14). Each node specifies its clinical context, trigger condition, diagnostic state, and termination flag, outlining the structured reasoning path from symptoms to confirmed diagnosis..

**Faithfulness**: Accurately encodes ADA diagnostic criteria, including FPG ($\geq$126 mg/dL), OGTT 2h-PG ($\geq$200 mg/dL), A1C ($\geq$6.5%), and random plasma glucose ($\geq$200 mg/dL) with symptoms or hyperglycemic crisis. Confirmatory testing logic (Nodes 6–9) is preserved. Minor redundancy exists in OGTT-related triggers (Nodes 3 and 12).

**Logic Coherence**: Node transitions are logically consistent and lead to appropriate terminal outcomes. Terminal states include both diagnostic conclusions (Nodes 5, 7, 9) and clinical actions (Node 14). Pre-test carbohydrate intake branches (Nodes 10–12) introduce unnecessary complexity.

**Clinical Utility**: Covers all major diagnostic pathways, including glucose thresholds, symptom-based criteria, and autoantibody testing for type 1 diabetes. Handles indeterminate or discordant results through non-terminal states (e.g., `ConflictResults`, `Unresolved`). High-risk clinical conditions (e.g., HIV, dialysis, transfusion) appear in the symptom set but are not explicitly integrated into decision pathways.

**Assessment**: The CEG exhibits high fidelity to guidelines and strong logic coherence. Recommended refinements include reducing redundancy, simplifying OGTT-related pathways, and integrating high-risk conditions into decision logic.

### C.4.3 HUMAN EVALUATION ON DIALOGUES AND CASE STUDY

| Turn | Dialogue (EHR + DFA Reasoning) |
|---|---|
| $t = 1$ | **Patient:** I'm a woman, not Hispanic. Lately I've been urinating a lot, feeling very thirsty, having blurry vision, and losing weight without trying
**AutoClinician:** Based on your reported symptoms, the current state is EvidenceGathering, and the next step is to conduct diagnostic tests such as FPG, OGTT, or A1C to confirm the diagnosis of diabetes.. |
| $t = 2$ | **Patient:** Here are my Lab results: Fasting plasma glucose=104.0 mg/dL; Glucose [Mass/volume] in Serum or Plasma (random)=180.0 mg/dL; Hemoglobin A1c/Hemoglobin.total in Blood=6.7 %;
**AutoClinician:** Based on the lab results, the patient has conflicting indicators for diabetes, with a fasting plasma glucose within normal range but elevated random glucose and hemoglobin A1c. Further testing such as a repeat A1C or an OGTT is needed to resolve the diagnostic uncertainty. |
| $t = 3$ | **Patient:** My A1C is 5.4%.
**AutoClinician:** Your A1C is below threshold, which conflicts with your FPG. Let's repeat the A1c or OGTT. |
| $t = 4$ | **Patient:** My OGTT fasting is 118 mg/dL.
**AutoClinician:** Repeat testing confirms abnormal OGTT. I have collect enough evidence and confirm you the E11 Type 2 diabetes mellitus |

Table 6: An EHR-driven Dialogue based on DFA.

We collaborated with clinicians to review approximately 30 generated dialogues and selected one diabetes-related case as a representative example. Human experts then evaluated the dialogue in Table 6 along three dimensions: guideline faithfulness, logic coherence, and clinical utility.

**Faithfulness to Guidelines**: Autoclinician correctly mapped patient-reported classic symptoms (polyuria, polydipsia, blurry vision, unintentional weight loss) to the EvidenceGathering state and recommended guideline-concordant diagnostic tests (FPG, OGTT, A1C). It appropriately interpreted numerical values: for example, FPG of 104 mg/dL was recognized as within the non-diabetic range (<126 mg/dL), random glucose of 180 mg/dL as elevated but below the diagnostic threshold (200 mg/dL), and A1C of 6.7% as above the diagnostic cutoff (6.5%). The model advised repeat testing when discordant results were observed, consistent with ADA recommendations. A minor inaccuracy occurred at Turn 3, where Autoclinician attributed the

conflict to FPG instead of the discrepancy between the initial elevated A1C (6.7%) and the subsequent normal A1C (5.4%).

**Logic Coherence**: Dialogue progression followed the deterministic DFA policy. Each transition was grounded in the observed inputs, and the terminal state ConfirmedDiagnosis (Type 2 diabetes) was reached only after confirmatory evidence had been obtained.

**Clinical Utility**: Autoclinician provided clear next steps at each turn, such as recommending repeat A1C or OGTT to resolve uncertainty, reflecting good alignment with guideline-directed clinical workflows. By explicitly interpreting laboratory values relative to diagnostic thresholds, the dialogue enhanced transparency and patient understanding. The final diagnosis of type 2 diabetes was appropriate. The explicit statement of diagnostic thresholds (e.g., OGTT is abnormal) further highlighted the model's interpretability and clinical credibility by clarifying normal and abnormal ranges

**Overall Assessment**. Autoclinician demonstrated high fidelity to guidelines, accurate interpretation of numerical test results, and appropriate use of guideline-based recommendations for further laboratory testing.

## C.5 SENSITIVITY ANALYSIS ON DIFFERENT JUDGE THRESHOLD

Table 7: Performance on Varying the LLM-as-a-judge Alignment Thresholds

| Dataset | 0.5 | 0.6 | 0.7 | 0.8 | 0.9 |
|---|---|---|---|---|---|
| EHRSHOT-Metabolic | 25.33 | 26.35 | 26.12 | 26.54 | 26.32 |
| EHRSHOT-Respiratory | 27.95 | 29.10 | 30.42 | 29.55 | 30.47 |
| EHRSHOT-Circulatory | 27.06 | 28.54 | 29.37 | 30.68 | 30.51 |

Table 7 Results report performance when varying the LLM-judge alignment threshold from 0.5 to 0.9. Across all disease settings, accuracy improves notably when increasing the threshold from 0.5 to 0.7, indicating that pseudo-patient examples effectively correct CEG errors and strengthen reasoning quality. Beyond 0.7, performance stabilizes and changes become marginal, demonstrating robustness to threshold choice. In our experiment, we adopt a slightly conservative threshold of 0.8.

## C.6 ABLATION STUDIES ON DIFFERENT PSEUDO-PATIENT TYPES.

Table 8: Ablation Study on Removing Different Refinement Types.

| Dataset | w/o Incomplete Evidence | w/o Threshold Misalignment | w/o Logic Errors | w/o Conflict Resolution Failure |
|---|---|---|---|---|
| EHRSHOT-Metabolic | 24.31 | 25.67 | 26.05 | 26.13 |
| EHRSHOT-Respiratory | 27.39 | 28.35 | 28.79 | 27.67 |
| EHRSHOT-Circulatory | 28.76 | 29.58 | 28.17 | 29.34 |

Table 8 presents ablations isolating each refinement component. Removing Incomplete Evidence correction leads to the largest degradation, reflecting the importance of capturing missing diagnostic signals. Threshold Misalignment removal also produces a measurable drop, especially in circulatory disorders where diagnostic thresholds are critical. Removing Logic Ordering or Conflict Resolution yields smaller but consistent declines, suggesting their contribution mainly affects ambiguous cases.

## C.7 Human Inter-rater Reliability

Table 9: Clinician Evaluation Cohen's Scores.

| Dimension | Mean Score (Clinician A) | Mean Score (Clinician B) | Cohen's $\kappa$ |
|---|---|---|---|
| Guideline Faithfulness | 4.6 | 4.8 | 0.74 |
| Logical Coherence | 4.4 | 4.5 | 0.69 |
| Clinical Utility | 3.3 | 3.1 | 0.65 |

We incorporated quantitative blinded human evaluation results. Two licensed clinicians independently reviewed 30 randomly sampled model-generated diagnostic dialogues. Each dialogue was rated using a 5-point Likert scale across three dimensions: guideline faithfulness, reasoning coherence, and clinical utility, where 1 indicates clinically incorrect reasoning and 5 represents excellent guideline concordance and interpretability. Experimental results are shown in Table 9.

To assess the consistency of the evaluation, we computed inter-rater agreement using Cohen's Cohen (1968). The resulting values indicate moderate to substantial agreement, demonstrating that the evaluation is reliable and not dependent on a single reviewer's subjective judgement. Importantly, clinicians noted that when evidence was insufficient, our model appropriately declined to make a diagnosis and switch to the base model rather than hallucinating or generating unsafe recommendations.

## C.8 Sensitivity to Different Judge Models.

Table 10: Performance Comparison on Different Judge Models.

| Model | EHRSHOT-Metabolic | | | EHRSHOT-Respiratory | | | EHRSHOT-Circulatory | | |
|---|---|---|---|---|---|---|---|---|---|
| | Acc | MRR | Lab Acc | Acc | MRR | Lab Acc | Acc | MRR | Lab Acc |
| QwQ-32B | 25.45 | 26.18 | 22.48 | 28.62 | 30.05 | 25.16 | 29.41 | 28.95 | 23.87 |
| DiagnosisGPT-34B | 26.10 | 26.75 | 23.01 | 29.20 | 30.66 | 26.81 | 30.21 | 29.03 | 24.65 |
| Qwen-32B | 26.54 | 26.92 | 23.37 | 29.55 | 31.09 | 27.32 | 30.68 | 29.14 | 24.98 |

To assess whether the judge model affects refinement outcomes, we evaluated AutoClinician with multiple LLM judges (Qwen-32B (we used in the paper), QwQ-32B, DiagnosisGPT-34B). As shown in Table 10, results remain highly consistent across judge model choices.

## C.9 Analysis on Fallback Rate

Table 11: Fallback Rate across Different Disease Subtypes.

| Category | Subtypes & Fallback Rates | | | |
|---|---|---|---|---|
| **Metabolic** | Thyroid (34%) | Diabetes (21%) | Metabolism (46%) | Endocrine (24%) |
| **Circulatory** | Hypertension (17%) | Coronary (26%) | Cerebrovascular (32%) | Aortic (43%) |
| **Respiratory** | Lower Respiratory Infection (12%) | Chronic Airway (19%) | Lung Injury (31%) | |

To quantify how often fallback occurs, we measured fallback frequency across disease subdomains, with results shown in Table 11. The fallback rate ranges from 12% to 46%, and this variability is clinically interpretable. The primary contributing factor is missingness in real EHR data. In real-world clinical settings,

patient observations (e.g., laboratory values) are often absent, either due to record errors or because tests were not ordered for clinical or cost-based reasons. In this work, we preserved data missingness rather than filtering incomplete samples, ensuring that evaluation reflects realistic deployment conditions. Our results demonstrate that guideline-grounded reasoning can be automated and applied to real EHRs. However, handling incomplete evidence and enabling trustworthy clinical reasoning under uncertainty remains an open and important direction for future research.

## C.10 RESULTS OF DIFFERENT OPEN-SOURCE BASE MODELS FOR CEG GENERATION AND REFINEMENT

Table 12: Results of Different Open-source Base model for CEG generation and Refinement.

| Model | EHRSHOT-Metabolic | | | EHRSHOT-Respiratory | | | EHRSHOT-Circulatory | | |
|---|---|---|---|---|---|---|---|---|---|
| | Acc | MRR | LabAcc | Acc | MRR | LabAcc | Acc | MRR | LabAcc |
| Yi-34B | 25.60 | 26.75 | 23.19 | 28.71 | 31.31 | 26.53 | 29.75 | 29.25 | 24.28 |
| GPT-OSS-120B | 27.11 | 27.25 | 24.56 | 30.22 | 32.87 | 28.11 | 30.50 | 30.47 | 25.50 |
| Qwen-32B | 26.54 | 26.92 | 23.37 | 29.55 | 31.09 | 27.32 | 30.68 | 29.14 | 24.98 |

To assess scalability and accessibility, we further tested refinement using Yi-34B and GPT-OSS-120B. As shown in Table 12, the resulting performance remains consistent across base models. We observed that GPT-OSS-120B tends to generate longer responses and higher inference latency. In contrast, Qwen-32B and Yi-34B produce cleaner structured reasoning and more accurate extraction of guideline logic, making them well-suited for CEG refinement and reasoning.

## C.11 STATISTICAL SIGNIFICANCE

Table 13: Statistically Significance.

| Dataset | Mean Accuracy (Baseline) | Mean Accuracy (AutoClinician) | Mean $\Delta$ | p-value |
|---|---|---|---|---|
| EHRSHOT-Metabolic | $22.88 \pm 0.12$ | $26.50 \pm 0.07$ | +3.62 | 0.0022 |
| EHRSHOT-Respiratory | $24.31 \pm 0.11$ | $29.00 \pm 0.13$ | +4.75 | 0.0009 |
| EHRSHOT-Circulatory | $22.47 \pm 0.14$ | $28.70 \pm 0.17$ | +6.21 | 0.0004 |

To evaluate whether the improvements introduced by AutoClinician are statistically significant rather than random variation, we conducted a paired t-test comparing our method against the strongest baseline (RAG-Qwen-32B w/ Query Planning) across the 5 runs. These results indicate that the improvement is highly statistically significant. The summary is Table 13.

AutoClinician consistently improves accuracy across all three datasets, with p-values $< 0.01$, indicating the gains are statistically significant rather than random variation. This confirms that AutoClinician's gains are robust, reliable, and not attributable to random noise.

