# OpenReview forum: "AutoClinician: Structured Clinical Guideline Integration for Trustworthy Diagnostic Reasoning in Healthcare"
_ICLR.cc/2026/Conference — Submitted to ICLR 2026_

### Official Review · Reviewer_rroq · 2025-10-27

**Soundness:** 2
**Presentation:** 2
**Contribution:** 2
**Rating:** 4
**Confidence:** 3

**Summary:**

The authors propose AutoClinician as a method for aligning LLMs with practical clinical workflows that can integrate existing medical guidelines. The paper demonstrates that AutoClinician can outperform alternative baseline methods by first turning narrative medical guidelines into structured ``Clinical Evidence Graphs'' (CEGs). It then tests and refines these graphs for accuracy using a self-consistency protocol without needing constant expert supervision. When diagnosing a patient the system uses these graphs to guide its reasoning tracking the conversation with a deterministic finite automaton to ensure logic stays on track. This means it is better at recommending the right lab tests and making an accurate final diagnosis compared to models that are just given raw guideline text or no guidance at all

**Strengths:**

* The methodology is an interesting and novel solution to a generally well-motivated problem
* The writing in the paper is written to a generally good standard
* Evaluation procedures are convincing in that every step of the methodology are essential particularly through the ablation studies

**Weaknesses:**

* **Consideration for data privacy in healthcare**: If I have understood correctly, the framework's graph generation relies on GPT-4o, a proprietary model accessible only through a third party API. Healthcare institutions handle very highly sensitive patient data and operate under strict privacy regulations. In my opinion, the need to send clinical guideline data (and potentially patient data for future applications) to an external service creates a significant barrier to real world adoption. This point is not discussed within the paper. Analyzing the performance of generating the graphs with open-source (and practical) LLMs is not explore.
* **Practicallity of real-world deployment**: The paper explores using open source models such as Qwen3-32B, but the overall diagnostic accuracy reported is very low. This level of accuracy is obviously insufficient for a clinical diagnostic tool where errors could have serious consequences for patient safety. The paper does not justify if this performance level is acceptable for practical implementation, which I believe is essential for the paper
* **Technical novelty**: the methodology, whilst sound, does not present much technical sophistication or novelty. I'm not sure it passes the high bar of ICLR

Minor:
* authors do not provide any analysis of the statistical significance of their method. Results in the main tables appear to be from a single run with no indication of variability, for example standard deviations from multiple runs with different seeds. This makes it difficult to assess the robustness and reliability of the findings
* The text in Figure 1 is too small

**Questions:**

* If a hospital cannot use a third-party API (e.g, due to data privacy concerns), can your pipeline generate CEGs with an on-prem open-source LLM? Did you try this in practice? If not explored, why not? What performance delta would you expect vs GPT-4o, and what evidence supports that expectation?
* The diagnostic accuracy with open source backbones looks very low. What minimum clinical threshold are you targeting, and how far are you from it?

---

> ### Author Response · Authors · 2025-11-26
> **Author Response to Reviewer  rroq**
>
> Thank you for your thoughtful review and constructive feedback! We address your concerns and answer your questions below. We also uploaded a revision and used blue to mark the new changes.
>
> **[Q1]: If a hospital cannot use a third-party API (e.g, due to data privacy concerns), can your pipeline generate CEGs with an on-prem open-source LLM? Did you try this in practice? If not explored, why not? What performance delta would you expect vs GPT-4o, and what evidence supports that expectation?**
>
> [A1]: **Clarification of Data Scope**. All experiments in this work were conducted exclusively using fully de-identified, publicly available research datasets. No protected health information or institution-specific private records were used at any stage of model development, refinement, or evaluation. In addition, all clinical guidelines were sourced from publicly accessible, non-restricted guideline repositories and government agencies.
>
> **Private Data Privacy in a Real Scenario**. Regarding deployment feasibility in real clinical environments, our method is compatible with privacy-preserving infrastructures. Many healthcare institutions, including Stanford Health Care, already support secure internal deployments of LLMs. For example, closed models such as GPT-4o are accessible through a secured hospital-controlled environment called SecureGPT (https://securegpt.stanfordhealthcare.org/chat). Researchers in their institution access these closed-source or open-source models using tools such as litellm. This demonstrates that both open-source and closed-sourced models can securely operate on private patient data.
>
> **[Q2]: The diagnostic accuracy with open source backbones looks very low. What minimum clinical threshold are you targeting, and how far are you from it?**
>
> [A2] Thank you for the thoughtful question. We clarify the interpretation of performance results from three perspectives:
>
> **(1) Task Difficulty and Realism.** Unlike prior work that evaluates models using simplified tasks such as multiple-choice QA or binary mortality prediction, our setting requires the model to reason over ~80 diagnosis categories based on heterogeneous and missing EHR observations and guidelines.
>
> While real-world usage may scope inference to narrower clinical areas (e.g., respiratory-only diagnosis prediction rather than full comprehensive reasoning), the full-domain setting provides a more stringent assessment of multi-condition medical reasoning. Under this challenging and realistic evaluation, AutoClinician consistently outperforms both general-purpose LLMs and medical domain models (see main results in Table 1 and Table 5), demonstrating measurable progress toward robust, guideline-aligned clinical reasoning.
>
> **(2) Beyond Prediction Accuracy**. In practical healthcare decision support, the value of AI systems is not only to provide a score, but require transparent reasoning, auditability, and alignment with guideline logic, not just accuracy scores. AutoClinician introduces a structured reasoning framework that extracts, verifies, and refines Clinical Evidence Graphs (CEGs) from authoritative guidelines, enabling interpretable and rule-consistent diagnostic reasoning rather than opaque prediction.
>
> **(3) Efforts Toward Clinical Trustworthiness Deployment**. Although current results is not able to be deployed in a real clinical scenario, our model can retrieve, align, and reason according to clinical guidelines automatically. This capability is essential for future reliable clinical AI systems, where explainability, correctness, and safety govern deployment viability.
> Overall, we view this work not as the final endpoint, but as a foundation toward trustworthy guideline-grounded reasoning systems, providing both measurable improvements and clinically meaningful interpretability advantages.
>
>
> **[Q3]:  Authors do not provide any analysis of the statistical significance of their method.**
>
> [A3]: To evaluate whether the improvements introduced by AutoClinician are statistically significant rather than random variation, we conducted a paired t-test comparing our method against the strongest baseline (RAG-Qwen-32B w/ Query Planning) across the 5 runs. These results indicate that the improvement is highly statistically significant. The summary is shown below:
>
> | Dataset | Mean Accuracy (Baseline) | Mean Accuracy (AutoClinician) | Mean Δ | p-value |
> |---------|--------------------------|--------------------------------|--------|---------|
> | EHRSHOT-Metabolic     | 22.88 ± 0.12 | 26.50 ± 0.07 | +3.62 | 0.0022 |
> | EHRSHOT-Respiratory   | 24.31 ± 0.11 | 29.00 ± 0.13 | +4.75 | 0.0009 |
> | EHRSHOT-Circulatory   | 22.47 ± 0.14 | 28.70 ± 0.17 | +6.21 | 0.0004 |
>
>  AutoClinician consistently improves accuracy across all three datasets, with p-values < 0.01, indicating the gains are statistically significant rather than random variation. This confirms that AutoClinician’s gains are robust, reliable, and not attributable to random noise.

---

### Official Review · Reviewer_V8s7 · 2025-10-31

**Soundness:** 3
**Presentation:** 3
**Contribution:** 3
**Rating:** 4
**Confidence:** 3

**Summary:**

AutoClinician proposes a training-free framework to enhance the trustworthiness and interpretability of Large Language Models in EHR-driven diagnostic conversations by rigorously integrating clinical guidelines. The framework first compresses complex narrative guidelines into structured Clinical Evidence Graphs (CEGs), which are automatically verified and refined using consistency-based adversarial pseudo-patient examples to ensure accuracy and fidelity to clinical logic. Diagnostic reasoning is then modeled as a deterministic, multi-step process guided by a Deterministic Finite Automaton (DFA), which acts as a state tracker, retrieving relevant CEGs and interpreting EHR data (including numerical lab values) to guide the LLM's conversational output. Experiments on EHRSHOT and TriNetX demonstrate that AutoClinician significantly outperforms standard zero-shot, few-shot, and iterative RAG baselines across diagnosis accuracy and lab test recommendation metrics, validating the benefit of structured guideline integration and DFA-guided reasoning.

**Strengths:**

- The structured integration of clinical guidelines via Clinical Evidence Graphs (CEGs) and the novel use of a Deterministic Finite Automaton (DFA) to manage multi-step, evidence-grounded conversational state are highly original for aligning LLMs with real-world clinical workflows.
- The proposed consistency-based refinement strategy using four types of adversarial pseudo-patient examples is a strong methodological contribution for unsupervised validation and correction of LLM-generated knowledge graphs.
- Modeling the diagnostic process as a DFA provides a highly interpretable and explicit reasoning scaffold that directly addresses the need for transparency and evidence-based medicine (EBM) principles in clinical AI systems.
- The comprehensive experimental validation across two challenging EHR benchmarks (EHRSHOT and TriNetX) and multiple ablation studies convincingly demonstrates the empirical superiority of the framework over competitive LLM baselines.

**Weaknesses:**

- The reliance on GPT-4o for the critical steps of CEG extraction and refinement introduces a potential limitation regarding reproducibility and scalability, as these steps are not validated with open-source models.
- While the DFA provides a strong structure, the definition of the five input categories ($\Sigma$) seems somewhat manual, and the robustness of the LLM's abstraction layer to accurately categorize complex EHR inputs into these abstract classes is not fully demonstrated.
- The paper claims the CEGs are automatically refined, yet the process relies on an LLM-as-a-judge scoring and then in-context refinement using failed examples, which is a powerful but potentially resource-intensive LLM operation whose exact cost/speed tradeoff is not discussed.
- The DFA design currently limits reasoning paths based only on the five retrieved CEGs, which might restrict the model's ability to handle highly rare or complex cases requiring broader medical context or non-standard guidelines.
- Although the CEG structure handles logical dependencies, the current 5-tuple format appears limited in capturing complex multi-factorial interactions or parallel diagnostic pathways common in highly heterogeneous diseases.
- The evaluation metrics, while standard (Acc, MRR, Lab Acc), do not fully capture the quality of the conversational output beyond the final outcome, particularly how the DFA handles the *ConflictResults* state and subsequent re-reasoning in practice.
- The human evaluation provided in the appendix is based on only one diabetes-related case example, which is insufficient to generalize the claims regarding guideline faithfulness and clinical utility across the full range of diseases covered in the benchmark.

**Questions:**

- Can the authors quantify the complexity (e.g., average number of nodes, depth, branching factor) of the generated CEGs for the different disease subsets (Metabolic, Respiratory, Circulatory), and how does CEG complexity correlate with refinement success?
- Given the reliance on GPT-4o for CEG extraction, have the authors explored validating the CEG generation and refinement process using alternative strong open-source models (e.g., Qwen3-32B or Yi-34B) to ensure the scalability and accessibility of the crucial knowledge preparation step?
- Could the authors provide a more detailed breakdown of the *Failure* input category ($\Sigma$) usage during DFA navigation in the experiments, specifically quantifying how often the model transitions to *Unresolved* and relies on the LLM itself, and how this rate compares between the disease subsets?

---

> ### Author Response · Authors · 2025-11-26
> **Author Response to Reviewer V8s730**
>
> Thank you for your thoughtful review and constructive feedback! We address your concerns and answer your questions below. We also uploaded a revision and used blue to mark the new changes.
>
> **[Q1]: Can the authors quantify the complexity (e.g., average number of nodes, depth, branching factor) of the generated CEGs for the different disease subsets (Metabolic, Respiratory, Circulatory), and how does CEG complexity correlate with refinement success?**
>
> [A1]: Thank you for the question. We quantified the structural complexity of generated CEGs across the three diseases and analyzed its relationship with refinement effectiveness. As shown in Table below, complexity varies meaningfully by clinical area. Metabolic guidelines yield smaller and shallower structures, whereas circulatory and respiratory guidelines have deeper paths and larger branching factors due to repeated complex confirmation steps and severity stratification.
> We further observe a positive correlation between CEG complexity and refinement gains: metabolic CEGs show modest improvement, whereas circulatory and respiratory domains benefit more substantially. This indicates that refinement is especially valuable with complexity guidelines and provides the greatest benefit where guideline reasoning is inherently intricate.
>
> | Clinical Domain | Avg. Nodes | Branching Factor | Pre-Refine Alignment (%) | Post-Refine Alignment (%) |
> |-----------------|------------|------------------|---------------------------|----------------------------|
> | Metabolic       | 9.4        | 3.6              | 75.3                      | 84.8                       |
> | Circulatory     | 14.7       | 2.3              | 71.1                      | 80.3                       |
> | Respiratory     | 16.1       | 2.6              | 65.4                      | 77.4                       |
>
> **[Q2]: Given the reliance on GPT-4o for CEG extraction, have the authors explored validating the CEG generation and refinement process using alternative strong open-source models (e.g., Qwen3-32B or Yi-34B) to ensure the scalability and accessibility of the crucial knowledge preparation step?**
>
> [A2]: Thank you for the question. We clarify that GPT-4o is used only for the initial CEG construction stage, where guideline content may include tables, structured formats for multimodal reasoning. The subsequent refinement stage does not rely on GPT-4o and it is performed using the open-source Qwen-32B model and is fully model-agnostic.
> To assess scalability and accessibility, we further tested refinement using Yi-34B and GPT-OSS-120B. As shown in Table below, the resulting performance remains consistent across base models. We observed that GPT-OSS-120B tends to generate longer responses and higher inference latency. In contrast, Qwen-32B and Yi-34B produce cleaner structured reasoning and more accurate extraction of guideline logic, making them well-suited for CEG refinement and reasoning.
>
> | Model | EHRSHOT-  |  Metabolic|  |  EHRSHOT-| Respiratory |  | EHRSHOT- |  Circulatory|  |
> |-------|--------------|--------------|----------------|-----------------|--------------|----------------|-----------------|--------------|----------------|
> |       | Acc | MRR | LabAcc | Acc | MRR | LabAcc | Acc | MRR | LabAcc |
> | Yi-34B | 25.60 | 26.75 | 23.19 | 28.71 | 31.31 | 26.53 | 29.75 | 29.25 | 24.28 |
> | GPT-OSS-120B | 27.11 | 27.25 | 24.56 | 30.22 | 32.87 | 28.11 | 30.50 | 30.47 | 25.50 |
> | Qwen-32B | 26.54 | 26.92 | 23.37 | 29.55 | 31.09 | 27.32 | 30.68 | 29.14 | 24.98 |

---

> > ### Author Response · Authors · 2025-11-26
> > **Author Response to Reviewer V8s730 (part 2)**
> >
> > **[Q3]: Could the authors provide a more detailed breakdown of the Failure input category () usage during DFA navigation in the experiments, specifically quantifying how often the model transitions to Unresolved and relies on the LLM itself, and how this rate compares between the disease subsets?**
> >
> > [A3]: To quantify how often the system transitions into the Failure input state and relies on free LLM reasoning, we measured fallback frequency across disease subdomains (shown as Table above). The fallback rate ranges from 12% to 46%, and this variation is clinically interpretable.
> > We found that fallback behavior is primarily influenced by two factors:
> >
> > **Missingness in Real-World EHR Data**.  Many guideline-required clinical observations (such as laboratory values) are missing in EHRs. In this work, we preserved missing values rather than filtering incomplete samples, so that evaluation reflects realistic deployment settings. However, incomplete evidence may prevent deterministic traversal of the CEG pathway, resulting in Unresolved states and triggering fallback.
> >
> > **Disease-Specific Diagnostic Complexity**. Some diagnoses require multi-step confirmatory criteria or multimodal evidence. DFor example respiratory-related diagnosis.. These scenarios show higher fallback rates because successfully navigating their CEG pathways requires multiple dependent observations to be available simultaneously. In contrast, diseases characterized by simpler single-threshold criteria (such as hypertension or obesity) exhibit substantially lower fallback frequency.
> >
> > | Category | | Metabolic| | | Circulatory | | | | Respiratory | | |
> > |----------|-----------|---|---|---|-------------|---|---|---|------------|---|---|
> > | Subtype  | Thyroid | Diabetes | Metabolism | Endocrine | Hypertension | Coronary | Cerebrovascular | Aortic | Lower Respiratory Infection | Chronic Airway | Lung Injury |
> > | Fallback Rate | 34% | 21% | 46% | 24% | 17% | 26% | 32% | 43% | 12% | 19% | 31% |
> >
> > Overall, fallback is most strongly associated with the interaction between guideline-required evidence and the structure of real-world EHRs. Improving reasoning robustness under incomplete evidence represents an important future direction for enhancing system reliability in clinical deployment.

---

> > > ### Comment · Reviewer_V8s7 · 2025-11-27
> > >
> > > Thanks for your comprehensive rebuttal, I have raised my scores accordingly.

---

> > > > ### Author Response · Authors · 2025-12-03
> > > > **Response to Reviewer V8s7**
> > > >
> > > > Thank you for the thoughtful assessment and for raising your score. We appreciate your feedback and recognition of our contributions.

---

### Official Review · Reviewer_3sjF · 2025-11-01

**Soundness:** 3
**Presentation:** 3
**Contribution:** 2
**Rating:** 6
**Confidence:** 4

**Summary:**

This paper proposes AutoClinician, a structured, training-free clinical reasoning framework that integrates medical guidelines into a graph-based representation called the Clinical Evidence Graph (CEG). By converting guideline text into structured conditional logic and using a deterministic finite automaton (DFA) to track patient states, the system aims to achieve interpretable, evidence-aligned diagnostic reasoning.

**Strengths:**

1. The paper is well-written and conceptually clear, with figures that effectively illustrate how CEG and DFA mirror real clinical workflows.

2. The approach is novel in aligning LLM reasoning directly with medical guidelines, producing a structured and interpretable reasoning path rather than unconstrained text generation.

3. The multi-step data construction (guideline parsing, graph generation, consistency validation, and DFA inference) reflects solid engineering design and an appreciation for clinical process logic.

**Weaknesses:**

1. Despite being “training-free,” the method still heavily depends on large language models for guideline extraction, pseudo-patient synthesis, and consistency scoring, which may introduce bias and circular evaluation.

2. The evaluation relies largely on an LLM-as-judge setup over limited samples, lacking confidence intervals or inter-rater validation, making statistical robustness unclear.

3. The comparison set omits stronger structured baselines such as knowledge-graph or rule-based RAG methods, so it remains uncertain whether the improvement stems from the CEG structure or general engineering advantages.

**Questions:**

1. How consistent are the results across different judging models or threshold settings for pseudo-patient validation?

2. Could the authors clarify the scope and versioning of the guideline corpus—how broadly does it cover specialties, and how are updates or conflicts handled?

3. When all retrieved CEGs fail to match a case and the system falls back to free LLM reasoning, how is guideline alignment maintained, and how often does this fallback occur?

---

> ### Author Response · Authors · 2025-11-26
> **Author Response to Reviewer 3sjF (Part 1)**
>
> Thank you for your thoughtful review and constructive feedback! We address your concerns and answer your questions below. We also uploaded a revision and used blue to mark the new changes.
>
>
> **[Q1]: How consistent are the results across different judging models or threshold settings for pseudo-patient validation?**
>
> [A1]: Thank you for the insightful question. We evaluated sensitivity with different judging models or threshold settings for pseudo-patient validation.
>
> **Threshold sensitivity**. Below Results report performance when varying the LLM-judge alignment threshold from 0.5 to 0.9. Across all disease settings, accuracy improves notably when increasing the threshold from 0.5 to 0.7, indicating that pseudo-patient examples effectively correct CEG errors and strengthen reasoning quality. Beyond 0.7, performance stabilizes and changes become marginal, demonstrating robustness to threshold choice. Since pseudo-patient cases encode full clinical attribute sets, a threshold of 0.7 already represents a strict acceptance condition. We therefore adopt a slightly conservative threshold of 0.8 in all experiments.
>
> | Dataset                | 0.5   | 0.6   | 0.7   | 0.8   | 0.9   |
> |------------------------|-------|-------|-------|-------|-------|
> | EHRSHOT-Metabolic      | 25.33 | 26.35 | 26.12 | 26.54 | 26.32 |
> | EHRSHOT-Respiratory    | 27.95 | 29.10 | 30.42 | 29.55 | 30.47 |
> | EHRSHOT-Circulatory    | 28.06 | 28.54 | 29.37 | 30.68 | 30.51 |
>
> **Sensitivity to Different Judge Models**. To assess whether the judge model affects refinement outcomes, we evaluated AutoClinician with multiple LLM judges (Qwen-32B (we used in the paper), QwQ-32B, DiagnosisGPT-34B). As shown in Table below, results remain highly consistent across judge model choices.
>
> | Model             | EHRSHOT- | Metabolic        |            | EHRSHOT- | Respiratory       |            | **EHRSHOT-** | Circulatory     |            |
> |-------------------|-----------------------|------------|-----|--------------------------|------------|-------|--------------------------|------------|----|
> |                   | Acc                   | MRR        | Lab Acc    | Acc                      | MRR        | Lab Acc    | Acc                      | MRR        | Lab Acc
> | QwQ-32B           | 25.45                 | 26.18      | 22.48      | 28.62                    | 30.05      | 25.16      | 29.41                    | 28.95      | 23.87      |
> | DiagnosisGPT-34B  | 26.10                 | 26.75      | 23.01      | 29.20                    | 30.66      | 26.81      | 30.21                    | 29.03      | 24.65      |
> | Qwen-32B  | 26.54             | 26.92  | 23.37 | 29.55               | 31.09  | 27.32  | 30.68               | 29.14  | 24.98  |
>
> **[Q2]: Could the authors clarify the scope and versioning of the guideline corpus—how broadly does it cover specialties, and how are updates or conflicts handled?**
>
> [A2]:  **Scope and Versioning of the Guideline Corpus, and Handling Updates/Conflicts**. To ensure clinical correctness and guideline alignment, we curated 56 guidelines from authoritative medical organizations, including the American Diabetes Association (ADA), American Thyroid Association (ATA), American College of Cardiology/American Heart Association (ACC/AHA), American Association of Clinical Endocrinology (AACE), among others. To maintain consistency and contemporary clinical relevance, we restricted guideline sources to the latest publicly released versions from 2021–2025.
>
> **Handling Updates and Conflicts**. To understand how guideline discrepancies should be managed, we consulted collaborating clinicians. In clinical practice, guidelines function as evidence-based consensus documents, developed through large-scale validation and expert review. Therefore, major contradictions across high-authority guideline sources are uncommon, as guidelines must satisfy regulatory, legal, and clinical accountability standards. When inconsistencies do arise, they are typically resolved by prioritizing the most recent publication or clarifying scope differences, rather than altering diagnostic logic during CEG construction.
> Observed variability generally falls into two categories, and our solution are as follows:
>
>   **1. Version Updates Driven by New Evidence.**
> When updated guidance becomes available, our system prioritizes the latest official guideline (e.g., ADA, ASA). Updated CEGs replace older versions while maintaining provenance when necessary.
>
>   **2. Scope Differences Rather Than True Conflicts.**
> Many apparent inconsistencies reflect differing applicability (e.g., general adult population vs. pregnancy-specific criteria). To accommodate this, we generate separate CEGs annotated with metadata such as target population, care setting, and geographic relevance. During inference, patient demographics and EHR context are used to retrieve the appropriate CEG, ensuring reasoning remains aligned with the correct clinical scope.

---

> > ### Author Response · Authors · 2025-11-26
> > **Author Response to Reviewer 3sjF (Part 2)**
> >
> > **[Q3]: When all retrieved CEGs fail to match a case and the system falls back to free LLM reasoning, how is guideline alignment maintained, and how often does this fallback occur? (related to data distribution)**
> >
> > A3: To quantify how often fallback occurs, we measured fallback frequency across disease subdomains, with results shown in Table below. The fallback rate ranges from 12% to 46%, and this variability is clinically interpretable. The primary contributing factor is missingness in real EHR data. In real-world clinical settings, patient observations (e.g., laboratory values) are often absent, either due to record errors or because tests were not ordered for clinical or cost-based reasons.
> > In this work, we preserved data missingness rather than filtering incomplete samples, ensuring that evaluation reflects realistic deployment conditions. Our results demonstrate that guideline-grounded reasoning can be automated and applied to real EHRs. However, handling incomplete evidence and enabling trustworthy clinical reasoning under uncertainty remains an open and important direction for future research.
> >
> > | Category |  |Metabolic  |  |  |  | Circulatory |  |  | | Respiratory |  |
> > |----------|------------------------|--|--|--|---------------------------|--|--|--|---------------------------|--|--|
> > | Subtype  | Thyroid | Diabetes | Metabolism | Endocrine | Hypertension | Coronary Disease | Cerebrovascular | Aortic Disease | Lower Respiratory Infection | Chronic Airway Disease | Lung Injury |
> > | **Fallback Rate** | 34% | 21% | 46% | 24% | 17% | 26% | 32% | 43% | 12% | 19% | 31% |

---

### Official Review · Reviewer_qp3y · 2025-11-02

**Soundness:** 2
**Presentation:** 2
**Contribution:** 2
**Rating:** 4
**Confidence:** 4

**Summary:**

AutoClinician is a training-free framework that integrates official clinical guidelines into EHR-driven diagnostic conversations. It compresses guidelines into Clinical Evidence Graphs (CEGs), refines them via pseudo–patient consistency checks, and conducts stepwise, evidence-grounded reasoning with a Deterministic Finite Automaton (DFA). Across EHRSHOT and TriNetX, it outperforms zero-/few-shot baselines and RAG-with-planning on diagnosis, MRR, and lab-test recommendation, with improved interpretability and guideline adherence.

**Strengths:**

- Aligns LLM reasoning with real clinical workflows and EBM principles.
- Structured, guideline-derived CEGs capture thresholds and ordered test logic beyond plain text retrieval.
- Scalable, training-free refinement using pseudo–patient consistency checks.
- DFA-based state tracking yields transparent, auditable reasoning steps (missingness, conflict, confirmation).
- Consistent performance gains on two EHR benchmarks; ablations show CEGs and DFA both matter.

**Weaknesses:**

- Heavy reliance on LLMs for CEG extraction/refinement and judging; potential bias, limited human gold standards.
- Pseudo–patient checks may miss rare or subtle guideline nuances; coverage breadth unclear.
- Evaluation uses embedding-based matching; limited details on human IRR and exact-match metrics.
- DFA design may oversimplify complex workflows; limited sensitivity analyses (e.g., CEG count, thresholds).
- Generalization to multimorbidity, unit normalization, and broader diseases needs more evidence.

**Questions:**

- What guideline sources/versions and disease coverage are included? Any clinician audits or agreement metrics on CEG correctness?
- How sensitive are results to judge thresholds and pseudo–patient volume/types? Can you quantify CEG error reduction over refinement iterations?
- How do results vary with alternative DFA designs and retrieval counts beyond top-5?
- Can you add exact-match metrics and blinded human ratings with inter-rater reliability for dialogue faithfulness?
- Do RAG baselines access the same guideline corpus, and how are context/latency constraints handled for deployment?

---

> ### Author Response · Authors · 2025-11-26
> **Author Response to Reviewer qp3y**
>
> Thank you for your thoughtful review and constructive feedback! We address your concerns and answer your questions below. We also uploaded a revision and used blue to mark the new changes.
>
> **[Q1]: What guideline sources/versions and disease coverage are included?**
>
> [A1]: Our framework covers 79 diagnosis categories across three major clinical domains. In the metabolic domain (23 diagnose subtypes), we include thyroid disorders (3 subtypes), diabetes subtypes (4), nutrition and metabolic disorders (8), and endocrine functional disorders (8). The circulatory domain includes 27 diagnoses comprising hypertension-related conditions (6), ischemic and coronary artery disease (6), cerebrovascular and stroke-related disorders (8), and peripheral vascular/aortic diseases (7). The respiratory domain includes 29 diagnoses spanning lower respiratory tract infections and pneumonia (16), chronic airway diseases (10), and lung injury/interstitial lung disease (3).
>
> To ensure guideline faithfulness and clinical reliability, we curated 56 guidelines from authoritative organizations, including the American Diabetes Association (ADA), American Thyroid Association (ATA), American Association of Clinical Endocrinology (AACE), American College of Cardiology/American Heart Association (ACC/AHA), etc. To maintain consistency and clinical relevance, all guideline sources were restricted to the most recent versions published between 2021 and 2025.
>
> **[Q2]: How sensitive are results to judge thresholds and pseudo–patient volume/types? Can you quantify CEG error reduction over refinement iterations?**
>
> [A2]: Thank you for the insightful comments! We evaluated sensitivity across both the LLM-judge threshold and pseudo-patient refinement design.
>
> **Threshold sensitivity**. Below Results report performance when varying the LLM-judge alignment threshold from 0.5 to 0.9. Across all disease settings, accuracy improves notably when increasing the threshold from 0.5 to 0.7, indicating that pseudo-patient examples effectively correct CEG errors and strengthen reasoning quality. Beyond 0.7, performance stabilizes and changes become marginal, demonstrating robustness to threshold choice. In our experiment, we adopt a slightly conservative threshold of 0.8.
>
> | Dataset | 0.5 | 0.6 | 0.7 | 0.8 | 0.9 |
> |---------|-----|-----|-----|-----|-----|
> | EHRSHOT-Metabolic | 25.33 | 26.35 | 26.12 | 26.54 | 26.32 |
> | EHRSHOT-Respiratory | 27.95 | 29.10 | 30.42 | 29.55 | 30.47 |
> | EHRSHOT-Circulatory | 27.06 | 28.54 | 29.37 | 30.68 | 30.51 |
>
> **Refinement contribution**. Below Results present ablations isolating each refinement component. Removing Incomplete Evidence correction leads to the largest degradation, reflecting the importance of capturing missing diagnostic signals. Threshold Misalignment removal also produces a measurable drop, especially in circulatory disorders where diagnostic thresholds are critical. Removing Logic Ordering or Conflict Resolution yields smaller but consistent declines, suggesting their contribution mainly affects ambiguous cases.
>
> | Dataset | w/o Incomplete Evidence | w/o Threshold Misalignment | w/o Logic Errors | w/o Conflict Resolution Failure |
> |-----------------------|--------------------------|----------------------------|------------------|--------------------------------|
> | EHRSHOT-Metabolic  | 24.31 | 25.67 | 26.05 | 26.13 |
> | EHRSHOT-Respiratory | 27.39 | 28.35 | 28.79 | 27.67 |
> | EHRSHOT-Circulatory | 28.76 | 29.58 | 28.17 | 29.34 |
>
> Together, these analyses indicate that (1) refinement performance is stable across threshold configurations and (2) all four refinement mechanisms meaningfully contribute to CEG correctness, with Incomplete Evidence and Threshold alignment being the most influential.
>
> **[Q3]: Evaluation uses embedding-based matching and can you add exact-match metrics?**
>
> [A3]: In this work, we adopt embedding-based matching rather than strict exact-match classification (see Lines 697–701). This design choice is motivated by the nature of real-world EHR diagnosis labels: 1) Multiple coding systems. Diagnoses may be represented using different ontologies (SNOMED-CT, ICD-9/10), making exact text comparison unreliable across formats. 2) Variable surface forms. The same condition may appear as a parent category, subtype, code expansion, or synonym (e.g., Type 2 diabetes, E11.9, SNOMED 44054006), which are clinically equivalent but not string-identical (as we discussed in lines 705-717).
>
> For example, Type 2 Diabetes Mellitus may appear as ICD-10 E11, E11.9, or the SNOMED code 44054006. Although clinically identical, exact-match scoring would count these as mismatches.
> In contrast, embedding-based evaluation is the appropriate choice for this task, as it reflects clinically meaningful similarity across coding systems, whereas exact matching would misrepresent equivalent diagnoses as errors.

---

> ### Author Response · Authors · 2025-11-26
> **Author Response to Reviewer qp3y [Part 2]**
>
> **[Q4]: How do results vary with alternative DFA designs and retrieval counts beyond top-5?**
>
> [A4]: we additionally tested retrieval depths up to 10 candidate CEGs, complementing the 1–5 range reported in Table 4 (Lines 397–411).  Increasing retrieval depth allows the model to incorporate a wider range of relevant guidelines, which leads to performance improvements across all clinical domains. However, the performance gains are marginal and do not scale proportionally with retrieval depth.
>
> An important factor is the high degree of missingness in EHR data, which limits the ability to fully align retrieved guideline logic with available patient evidence. Although additional guidelines increase theoretical reasoning coverage, practical benefit depends on whether relevant clinical variables are present in the EHRs. In our evaluation setting, we do not remove samples or missing values in order to align with real data distribution. Considering both accuracy and inference efficiency, we adopt k=5 as the default CEGs numbers.
>
> | Dataset        | Acc@1 | Acc@3 | Acc@5 | Acc@10 |
> |--------------|-------|-------|-------|--------|
> | EHRSHOT-Metabolic    | 19.47 | 23.31 | 26.54 | 27.95  |
> | EHRSHOT-Respiratory  | 21.12 | 26.29 | 29.55 | 31.61  |
> | EHRSHOT-Circulatory  | 18.86 | 22.15 | 30.68 | 30.76  |
>
> **Q5: Can you add exact-match metrics and blinded human ratings with inter-rater reliability for dialogue faithfulness?**
>
> [A5]: We incorporated quantitative blinded human evaluation results. Two licensed clinicians independently reviewed 30 randomly sampled model-generated diagnostic dialogues (Case Study is shown in Appendix C4.2). Each dialogue was rated using a 5-point Likert scale across three dimensions: guideline faithfulness, reasoning coherence, and clinical utility, where 1 indicates clinically incorrect reasoning and 5 represents excellent guideline concordance and interpretability.
>
> To assess the consistency of the evaluation, we computed inter-rater agreement using Cohen’s κ [1]. The resulting values indicate moderate to substantial agreement, demonstrating that the evaluation is reliable and not dependent on a single reviewer’s subjective judgement. Importantly, clinicians noted that when evidence was insufficient, our model appropriately declined to make a diagnosis and switch to the base model rather than hallucinating or generating unsafe recommendations.
>
>
> | **Dimension**              | **Mean Score (Clinician A)** | **Mean Score (Clinician B)** | **Cohen’s κ** |
> |---------------------------|------------------------------|------------------------------|---------------|
> | Guideline Faithfulness    | 4.6                          | 4.8                          | 0.74          |
> | Logical Coherence         | 4.4                          | 4.5                          | 0.69          |
> | Clinical Utility          | 3.3                          | 3.1                          | 0.65          |
>
> Together, these quantitative results support that our models' reasoning is not only qualitatively guideline-aligned but also consistently judged as clinically meaningful by independent human experts.
>
> **[Q6]: Do RAG baselines access the same guideline corpus, and how are context/latency constraints handled for deployment?**
>
> [A6]: Yes. All RAG baselines in our experiments access the exact same guideline corpus as AutoClinician to ensure a fair  comparison. The key difference lies in how retrieved evidence is consumed. RAG systems retrieve guideline text chunks and pass the full passage to the model for reasoning, whereas AutoClinician retrieves only the corresponding CEGs, resulting in a more compact and interpretable reasoning input.
> Because the CEG-based input is substantially smaller than full guideline text, AutoClinician exhibits lower prompt length and reduced inference latency. The guideline preprocessing pipeline are detailed in Appendix (Lines 680–687).
>
>
> Reference:
>
> [1] Cohen J. Weighted kappa: Nominal scale agreement provision for scaled disagreement or partial credit. Psychological bulletin. 1968 Oct;70(4):213.

---

> > ### Comment · Reviewer_qp3y · 2025-11-27
> >
> > Thank you for your response, which has partially addressed my concerns. I am still not fully convinced of the work’s novelty, as its technical contribution seems incremental. The paper would benefit from clearly articulating any new phenomena, observations, or implementation‑related limitations, and from presenting motivations beyond benchmark gains to better demonstrate its broader impact.

---

> > > ### Author Response · Authors · 2025-12-03
> > > **Response to Reviewer qp3y**
> > >
> > > We appreciate your thoughtful comments and the opportunity to further clarify the novelty and contribution of our work.
> > >
> > > First, ***our work formulates a new problem setting***. Existing systems focus on clinical QA or supervised diagnosis prediction, which cannot guarantee trustworthy, guideline-aligned reasoning in real diagnostic workflows. To address this fundamental gap, we study how to ***automatically operationalize clinical guidelines and align the model’s reasoning process with real clinical workflow rather than only improving benchmark scores***.
> > >
> > > Second, ***we report new empirical observations on guideline extraction and propose a solution not explored in prior literature***. Through collaboration with clinicians, we identify four common failure modes in LLM-based extraction. These findings motivate our self-extraction and self-refinement pipeline, which converts long, heterogeneous guidelines into validated Clinical Evidence Graphs without human annotation.
> > >
> > > Third, ***we introduce a deterministic finite-automaton controller that enforces state-consistent reasoning in multi-turn diagnostic dialogue***. Prior LLM systems generate unconstrained reasoning steps without guarantees of clinical validity. Our design synchronizes each reasoning transition with guideline logic, producing auditable and safety-aligned diagnostic trajectories.
> > >
> > > Together, these contributions directly target the central challenge of ***ensuring trustworthiness in clinical diagnostic models***. They represent a new paradigm for guideline-grounded medical agents rather than an incremental extension of existing approaches. ***To support this, we curated 56 authoritative guidelines covering 79 diagnostic categories across three major clinical domains.*** We will revise the manuscript to more clearly highlight the new observations, methodological innovations, and their broader implications for clinically reliable AI systems.

---

### Author Response · Authors · 2025-12-03
**General Response to all reviewers**

We would like to sincerely thank the reviewers for their thoughtful feedback. In particular, we appreciate Reviewer 3sjF for recognizing the novelty of our structured and interpretable reasoning paths, and Reviewer V8s7 for raising the score and highlighting the originality of our Clinical Evidence Graph framework and DFA-based conversational controller. Below is the summary of new results and updates made based on feedback from reviewers:

- ***Sensitivity Analyses Added***: We add sensitivity studies for both LLM-judge decision thresholds and the pseudo-patient refinement types, demonstrating consistent performance across varying configurations.


- ***Extended Retrieval Depth Evaluation***: Retrieval experiments now include candidate Clinical Evidence Graphs (CEGs) up to a depth of 10, complementary to the depth range of 1–5 in our initial manuscript.


- ***Blinded Expert Assessment***: We introduced blinded human evaluation and reported inter-rater reliability (Cohen’s κ), confirming strong expert consistency and validating qualitative findings.


- ***Fallback Behavior Quantified***: We quantified fallback frequency across clinical subdomains to provide clearer interpretability and error pattern analysis.


- ***Structural Complexity Analysis***: We measured and reported CEG structural properties (e.g., number of nodes, depth, branching factor) across three disease subdomains and analyzed their correlation with refinement effectiveness.


- ***Model Generalization Validation***: To evaluate generalizability, we added experiments using two additional strong open-source models: DiagnosisGPT-34B and GPT-OSS-120B, demonstrating consistent trend.


- ***Statistical Robustness***: We conducts five independent runs with statistical significance testing to ensure robustness and reproducibility.

We discuss more about our contribution and novelty here, which is concerned by Reviewer qp3y:

1. ***Previous methods are limited in providing trustworthy clinical reasoning evidence.***
Existing systems focus on clinical QA or purely diagnosis prediction and therefore cannot guarantee trustworthy, guideline-aligned reasoning in real workflows. Our work addresses this gap by enabling fully automated alignment of model reasoning process with authoritative clinical guidelines.  To support this, we curate 56 guideline documents spanning 79 diagnostic categories across major clinical domains and demonstrate that our framework delivers consistently more reliable, guideline-grounded reasoning than both general-purpose and medical-specific LLM baselines.

2. ***Our method shows strengths in automatically extracting and refining evidence from guidelines without annotation.***
Clinical guidelines are long, heterogeneous, and logic-dense, making annotation-based efforts impractical. Through collaboration with clinicians, we identify four common failure modes in LLM-based extraction and introduce a fully automated self-extraction + self-refinement pipeline that transforms raw guideline text into validated Clinical Evidence Graphs without human labeling.

3. ***our methods show advantages in aligning diagnosis  reasoning process with clinical logics.***  To prevent unconstrained generation and enforce clinically valid reasoning steps, we design a DFA-based controller that synchronizes dialogue progression with guideline logic. This enables auditable, step-wise state transitions that preserve safety, consistency, and diagnostic validity.

---

### Meta-Review · Area_Chair_bVsU · 2026-01-09

**Summary:**

While the paper presents a well-engineered and thoughtfully evaluated system and the rebuttal addressed many execution-level concerns (e.g., sensitivity analyses, human evaluation, statistical testing, and robustness checks), the proposed approach: guideline extraction into structured graphs, self-consistency refinement, and DFA-based control, mostly largely combine existing techniques (RAG, graph structuring, LLM-as-judge, constrained decoding) without introducing a fundamentally new learning principle or algorithm, a concern explicitly raised by  multiple reviewers.

**Reviewer Concerns:**

addressed many execution-level concerns (e.g., sensitivity analyses, human evaluation, statistical testing, and robustness checks

**Reviewer Scores:**

unchanged

---

### Decision · Program_Chairs · 2026-01-26

Reject